# ONE-FOR-ALL FEW-SHOT ANOMALY DETECTION VIA INSTANCE-INDUCED PROMPT LEARNING

**Wenxi Lv**[1], **Qinliang Su**[1,2]*, **Wenchao Xu**[3]

[1] School of Computer Science and Engineering, Sun Yat-sen University, Guangzhou, China.
[2] Guangdong Key Laboratory of Big Data Analysis and Processing, Guangzhou, China.
[3] Department of Computing, The Hong Kong Polytechnic University, Hong Kong SAR

`lvwx8@mail2.sysu.edu.cn`      `suqliang@mail.sysu.edu.cn`
`wenchao.xu@polyu.edu.hk`

## ABSTRACT

Anomaly detection methods under the 'one-for-all' paradigm aim to develop a unified model capable of detecting anomalies across multiple classes. However, these approaches typically require a large number of normal samples for model training, which may not always be fulfilled in practice. Few-shot anomaly detection methods can address scenarios with limited data but require a tailored model for each class, following the 'one-for-one' paradigm. In this paper, we first proposed a one-for-all few-shot anomaly detection method with the assistance of vision-language models. Unlike previous CLIP-based methods that learn fixed prompts for each class, our method learns a class-shared prompt generator to adaptively generate suitable prompts for each instance. The prompt generator is trained by aligning the prompts with the visual space and utilizing guidance from general textual descriptions of normality and abnormality. In addition, we further propose a method to address the problem of how to retrieve valid similar features from the visual memory bank under the one-for-all paradigm. Extensive experimental results on MVTec and VisA demonstrate the superiority of our method in few-shot anomaly detection task under the one-for-all paradigm. Our code is available in `https://github.com/Vanssssry/One-For-All-Few-Shot-Anomaly-Detection`.

## 1 INTRODUCTION

Visual anomaly detection aims to detect anomalies in images, which has widespread applications across various fields like industrial damage inspection (Bergmann et al., 2019; Zou et al., 2022), medical diagnosis (Zhang et al., 2020; Fang et al., 2024), *etc.*. Due to the countless forms and types of potential anomalies in real-world applications, as well as the difficulties of collecting anomalies, existing anomaly detection methods (Deng & Li, 2022; Tien et al., 2023; Roth et al., 2022; Liu et al., 2023; Schlüter et al., 2022) mostly assume the only availability of normal samples during training and then use them to train a normality model that can capture any samples that deviate from the normality. Despite achieving high detection accuracy, these methods follow the 'one-for-one' paradigm, which requires a bespoke anomaly detection model to be trained for each class in the dataset, which, obviously, is cumbersome to use and expensive to train. To increase the flexibility and reduce the training complexity, some recent works have proposed to resort to the 'one-for-all' paradigm as proposed in UniAD in (You et al., 2022), DiAD in (He et al., 2024), and VPDM in (Li et al., 2024c). Different from the 'one-for-one' paradigm, methods under the 'one-for-all' paradigm only need to train a common model for all classes of data, and thus could significantly increase the using flexibility and training efficiency.

However, all of the 'one-for-one' and 'one-for-all' methods above assume the availability of a large amount of normal instances for model training. But for some application scenarios, this requirement may not be easy to meet, *e.g.*, at the initial phase of manufacturing a new product or in scenarios with difficulties for instance collection. To deal with these scenarios, few-shot anomaly detectors

---
*Corresponding author.

have been proposed, such as RegAD in (Huang et al., 2022) and FastRecon in (Fang et al., 2023), which use several available images to estimate the distribution of normal instances or use them to build a coreset for instance reconstruction. But due to the scarcity of training data, the distribution or coreset estimated by these methods is very inaccurate, causing their detection performance to lag far behind their full-shot counterparts. Recently, inspired by advancements of vision-language models (VLMs) (Radford et al., 2021; Zhou et al., 2022b), WinCLIP (Jeong et al., 2023) proposed to manually craft a set of prompts that describe the general properties of various anomalies and then use CLIP to assess the alignment between image patches and any of the anomaly prompts. Patches with high anomaly alignment scores are considered anomalous. However, as CLIP struggles with capturing local information, WinCLIP generates hundreds of image windows of different sizes and employs CLIP's visual encoder to extract their features, which obviously is computationally expensive. Moreover, the diversity and complexity of anomalies make it impossible to have manually crafted prompts cover or accurately describe all possible types of anomalies. To address the two issues, the recent work PromptAD (Li et al., 2024b) employs the V-V attention mechanism (Li et al., 2023b) to preserve local information in the output tokens of CLIP and then further proposes to learn a set of vectors (prompts) from data automatically. Despite remarkable performance improvement and significant complexity reduction, PromptAD, as well as WinCLIP, still fall within the 'one-for-one' paradigm. That is, it requires to learn a separate prompt for every category of data, which could incur a huge computation burden for training and cause inconvenience for practical utilization.

In this paper, we propose the first one-for-all few-shot anomaly detection method by harnessing the zero/few-shot recognition ability of vision-language models CLIP (Radford et al., 2021) and BLIP-Diffusion (Li et al., 2024a) via instance-induced prompt learning, naming our method instance-induced prompt anomaly detection (IIPAD). Specifically, instead of learning a fixed prompt for every category as in WinCLIP and PromptAD, we propose to learn a common prompt generator for all categories. The prompt-generator is designed to assemble a common sequence of vectors and a set of instance-specific tokens output from CLIP and Q-Former in BLIP-Diffusion (Li et al., 2024a) via concatenation and addition. In this way, the prompt generator can generate an exclusive prompt for every instance by taking its own characteristics into account, making the prompts better at extracting instance-specific normal and abnormal details. Given the designed prompt generator, the model is trained by seeking to align the prompt and visual embeddings corresponding to the same state (normal vs abnormal). The cross-modal alignment is further conducted at the token level to strengthen the model's fine-grained understanding ability. In addition to the prompt-visual alignment, we also introduce a set of general textual descriptions about normal and abnormal states, which can be viewed as a kind of expert knowledge, and use them to guide the training process by encouraging prompts to align with them. Furthermore, the memory bank is adopted within the one-for-all paradigm by endowing it with category-aware capabilities. Extensive experimental results on MVTec and VisA datasets demonstrate that our proposed one-for-all few-shot anomaly detection method shows significantly better performance than baselines under the one-for-all setting, and can achieve similar or even better performance than the SOTA one-for-one few-shot method.

## 2 RELATED WORK

**Few-Shot Anomaly Detection**  Few-shot anomaly detection is designed for scenarios where only a limited amount of normal data is available for training. TDG (Sheynin et al., 2021) proposed to augment images in the support set through various transformations and leverage a hierarchical generative model to learn the multi-scale patch distribution. Instead of image augmentation, RegAD (Huang et al., 2022) leverages an auxiliary dataset to learn the matching mechanism for anomaly detection on target dataset. DifferNet (Rudolph et al., 2021) leverages a normalizing flow to estimate the distribution of descriptive features extracted by a pre-trained model. FastRecon (Fang et al., 2023) and GraphCore (Xie et al., 2023) build on PatchCore (Roth et al., 2022). FastRecon learns a projection matrix to reconstruct features as normal, while GraphCore enlarges the normal feature bank through data augmentation and trains a graph neural network to identify anomalies. However, all these methods follow a "one-model-per-category" paradigm.

**Leveraging VLMs for Zero-/Few-shot Anomaly Detection**  Since CLIP (Radford et al., 2021) demonstrates remarkable performance in zero-shot and few-shot classification, it has gained significant attention and is widely studied for its potential application in anomaly detection. Win-

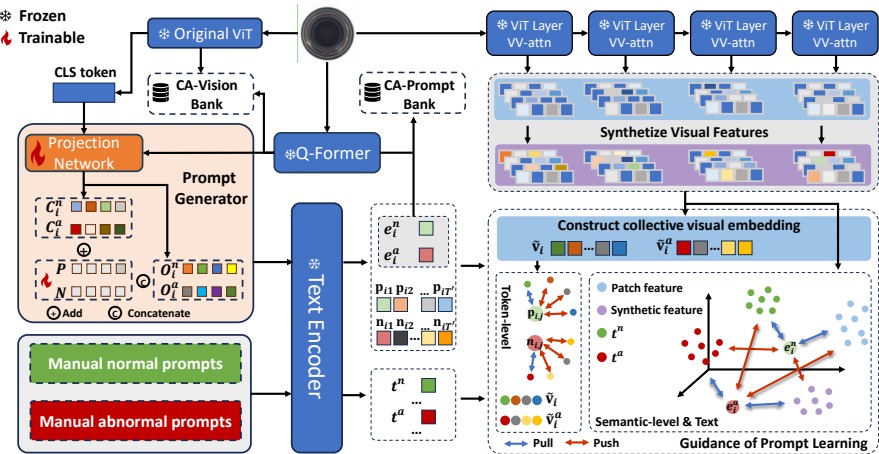

Figure 1: The framework of IIPAD. The Q-Former is leveraged to extract object tokens for prompt and category-aware memory bank construction. The prompt generator is trained using contrastive learning, guided by both visual and textual modalities, to produce instance-specific prompts for anomaly detection. The ViT used to extract patch features is transformed with V-V attention.

CLIP (Jeong et al., 2023) leverages manual text prompts to detect anomalies across predefined windows. AnoVL (Deng et al., 2023) uses an adapter to integrate text prompts with visual patches extracted through a V-V attention-based visual encoder. Getting rid of abundant manual text prompts, PromptAD (Li et al., 2024b) proposes to learn normal and anomalous prompts via semantic concatenation. But these methods still follow the one-for-one paradigm, learning tailored prompts for each class. AnomalyCLIP (Zhou et al., 2024) utilizes an auxiliary dataset from another domain to learn object-agnostic text prompts that capture generic normality and abnormality across categories, while AdaCLIP (Cao et al., 2024) uses the auxiliary dataset to learn hybrid prompts. InCTRL (Zhu & Pang, 2024) proposes leveraging a few normal images as prompts to conduct residual learning, which captures general abnormality for anomaly detection. However, AnomalyCLIP and AdaCLIP require thousands of normal and anomalous images with pixel-level annotations, while InCTRL struggles with the anomaly localization task.

## 3 METHODS

In the task of few-shot anomaly detection, it is assumed the availability of a training dataset, $\mathcal{X}_{train} = \{\mathbf{x}_1, \mathbf{x}_2, \ldots, \mathbf{x}_N\}$, consisting of a small number of normal images from multiple classes. Each class contains at most $K$ examples, where $K$ typically ranges from 1 to 4. In this paper, our goal is to leverage the small dataset $\mathcal{X}_{train}$ to learn a set of prompts to help detect and localize anomalies across various categories. To this end, we first develop a class-shared prompt generator, which can generate an instance-specific prompt for every instance. Then, we train the prompt generator by encouraging the alignment between the prompt and visual space as well as exploiting the guidance from general textual descriptions on normality and abnormality. Furthermore, a category-aware memory bank is further introduced to incorporate the test-time knowledge to detect anomalies. The framework of our method is shown in Figure 1.

### 3.1 INSTANCE-SPECIFIC PROMPT GENERATOR

To avoid manually crafting the prompts, inspired by CoOp (Zhou et al., 2022b), existing prompt-based anomaly detection methods like AnomalyCLIP and PromptAD can be described as introducing a set of learnable vectors for each category of data and then learning them separately. The prompts for the $c$-th category in these methods can be generally described

$$\mathbf{S}_c^n = [\mathbf{P}_{c1}][\mathbf{P}_{c2}]\ldots[\mathbf{P}_{cT}][\text{CLS}_c], \quad \mathbf{S}_c^a = [\mathbf{N}_{c1}][\mathbf{N}_{c2}]\ldots[\mathbf{N}_{cT}][\text{CLS}_c], \quad (1)$$

where $\mathbf{S}_c^n$ and $\mathbf{S}_c^a$ denote the normal and abnormal prompts for the $c$-th category; $[\mathbf{P}_{ci}]$ and $[\mathbf{N}_{ci}]$ denote the $i$-th token and $T$ is the total length; and $[\text{CLS}_c]$ represents the tokens of $c$-th class name. Obviously, this prompt design makes the method fall into the 'one-for-one' paradigm, which could cause significant computation burden for training and incur inconvenience for practical use.

To adapt these methods to the 'one-for-all' paradigm, a very simple way is to make the tokens $[\mathbf{P}_{ci}]$ and $[\mathbf{N}_{ci}]$ not reliant on the class $c$, leading to the prompts $\mathbf{S}_c^n = [\mathbf{P}_1][\mathbf{P}_2]\ldots[\mathbf{P}_T][\text{CLS}_c]$ and $\mathbf{S}_c^a = [\mathbf{N}_1][\mathbf{N}_2]\ldots[\mathbf{N}_T][\text{CLS}_c]$. Now, since the tokens $[\mathbf{P}_i]$ and $[\mathbf{N}_i]$ are shared among all instances, the only difference of prompts comes from the class name token $[\text{CLS}_c]$. However, the class name can only describe instances coarsely, resulting in these prompts not being expressive enough to capture the subtle difference between normal and abnormal instances. With the development of VLMs, an image can be described by several tokens that are extracted from VLMs (Gal et al., 2023; Li et al., 2023a; 2024a). Since the Q-Former in BLIP-Diffusion (Li et al., 2024a) is trained together with CLIP to output a set of tokens that not only describe the image content but also can be interpreted by the CLIP textual encoder, we here propose to use it to extract a set of object tokens $\mathbf{Z}_i \in \mathbb{R}^{M \times d}$ for the image $\mathbf{x}_i$. However, we note that directly using the object tokens $\mathbf{Z}_i$ to replace the class name $[CLS]$ does not work very well, which is probably due to the fact that VLMs are not trained to capture the subtle difference between normality and abnormality of images. Thus, we propose to first use two MLPs, $f_c^n$ and $f_c^a : \mathbb{R}^{M \times d} \to \mathbb{R}^{m \times d}$ ($m < M$), to transform $\mathbf{Z}_i$ into two distinct tokens $\mathbf{O}_i^n$ and $\mathbf{O}_i^a \in \mathbb{R}^{m \times d}$ that are responsible for describing the normality and abnormality of images, respectively. Then, we use $\mathbf{O}_i^n$ and $\mathbf{O}_i^a$ to design the normal and abnormal prompts as

$$\mathbf{S}_i^n = [\mathbf{P}_1][\mathbf{P}_2]\ldots[\mathbf{P}_T][\mathbf{O}_{i1}^n]\ldots[\mathbf{O}_{im}^n], \quad \mathbf{S}_i^a = [\mathbf{N}_1][\mathbf{N}_2]\ldots[\mathbf{N}_T][\mathbf{O}_{i1}^a]\ldots[\mathbf{O}_{im}^a]. \quad (2)$$

Since the prefixes $[\mathbf{P}_i]$ and $[\mathbf{N}_i]$ are still shared among all instances, these prompts are still restrictive in capturing the subtle normality and abnormality in different images. To address this issue, inspired by CoCoOp (Zhou et al., 2022a), we further propose to inject instance-specific visual information into the tokens $[\mathbf{P}_i]$ and $[\mathbf{N}_i]$. Specifically, for instance $\mathbf{x}_i$, we first propose a way to extract instance-specific visual normal and abnormal tokens $\mathbf{C}_i^n \in \mathbb{R}^{T \times d}$ and $\mathbf{C}_i^a \in \mathbb{R}^{T \times d}$ from the output tokens of CLIP and Q-Former. Then, inspired by CoCoOp (Zhou et al., 2022a), we add the visual normal and abnormal tokens into the prompts $[\mathbf{P}_i]$ and $[\mathbf{N}_i]$, leading to the final form

$$\begin{aligned}\mathbf{S}_i^n &= [\mathbf{P}_1 + \mathbf{C}_{i1}^n][\mathbf{P}_2 + \mathbf{C}_{i2}^n]\ldots[\mathbf{P}_T + \mathbf{C}_{iT}^n][\mathbf{O}_{i1}^n]\ldots[\mathbf{O}_{im}^n], \\ \mathbf{S}_i^a &= [\mathbf{N}_1 + \mathbf{C}_{i1}^a][\mathbf{N}_2 + \mathbf{C}_{i2}^a]\ldots[\mathbf{N}_T + \mathbf{C}_{iT}^a][\mathbf{O}_{i1}^a]\ldots[\mathbf{O}_{im}^a],\end{aligned} \quad (3)$$

where $\mathbf{C}_{it}^n$ denotes the $t$-th token of $\mathbf{C}_i^n$. Here, the learnable prefixes $[\mathbf{P}_i]$ and $[\mathbf{N}_i]$ are responsible to capture general descriptions of normality and abnormality, while $\mathbf{C}_i^n$ and $\mathbf{C}_i^a$ aim to provide normal and anomalous details tailored for a specific image. In this way, instance-specific prompts can be generated to detect anomalies, rather than relying on a prompt shared across all instances.

To obtain $\mathbf{C}_i^n$ and $\mathbf{C}_i^a$, we propose to simultaneously make use of the class token extracted from CLIP's visual encoder $E_v(\cdot)$ and the category tokens $\mathbf{Z}_i$ from Q-Former. The class token $E_v(\mathbf{x}_i)$ is not only projected into the textual token space via an MLP $f_p : \mathbb{R}^d \to \mathbb{R}^{T \times d}$ as done in CoCoOp, but is also leveraged as a query in cross-attention to retrieve information from category tokens $\mathbf{Z}_i$ as

$$\mathbf{C}_i^n = f_p(E_v(\mathbf{x}_i)) + f_n(\text{CA}(E_v(\mathbf{x}_i), \mathbf{Z}_i^n, \mathbf{Z}_i^n)), \quad \mathbf{C}_i^a = f_p(E_v(\mathbf{x}_i)) + f_a(\text{CA}(E_v(\mathbf{x}_i), \mathbf{Z}_i^a, \mathbf{Z}_i^a)), \quad (4)$$

where $\text{CA}(\cdot)$ denotes the cross-attention operation, while $f_n, f_a : \mathbb{R}^d \to \mathbb{R}^{T \times d}$ denote two MLPs; and $\mathbf{Z}_i^n$ and $\mathbf{Z}_i^a$ are obtained from $\mathbf{Z}_i$ via two linear projection. Utilizing the two sources of visual information encourages $\mathbf{C}_i^n$ and $\mathbf{C}_i^a$ to capture both global visual information from the class token and fine-grained details from the category tokens.

By passing the prompts $\mathbf{S}_i^n$ and $\mathbf{S}_i^a$ into the CLIP textual encoder $E_T(\cdot)$, we obtain

$$\mathbf{e}_i^n, \mathbf{p}_{i1}, \mathbf{p}_{i2}, \ldots, \mathbf{p}_{iT'} = E_T(\mathbf{S}_i^n), \quad \mathbf{e}_i^a, \mathbf{n}_{i1}, \mathbf{n}_{i2}, \ldots, \mathbf{n}_{iT'} = E_T(\mathbf{S}_i^a), \quad (5)$$

where $T' = T + m$ is the length of input sequence; $\mathbf{e}_i^n$ and $\mathbf{e}_i^a \in \mathbb{R}^d$ denote the prompt embeddings of $\mathbf{S}_i^n$ and $\mathbf{S}_i^n$, respectively; $\mathbf{p}_{ij}$ and $\mathbf{n}_{ij}$ denote token embeddings of $j$-th token in $\mathbf{S}_i^n$ and $\mathbf{S}_i^n$;

## 3.2 MULTI-MODAL PROMPT TRAINING

To enable the prompt generator to produce tailored normal and anomalous prompts for the corresponding input image, we propose guiding the prompt learning in both visual and textual modalities.

### 3.2.1 VISUAL GUIDANCE

Since we can only access a few normal samples, we apply contrastive learning to pull the normal prompt embedding closer to normal visual features, while pushing the anomalous prompt embedding away. This approach is consistent with the goal of CLIP-based methods to detect anomalies.

During training, considering the trade-off between utilizing multi-level semantic information and computational cost, we select the outputs from certain intermediate layers of the CLIP visual encoder, with the set of indices of selected layers denoted as $\mathcal{J}$. Then, we use the selected features to construct the set $\mathcal{F}_i = \{\mathbf{F}_{ij} | j \in \mathcal{J}\}$, where $\mathbf{F}_{ij} \in \mathbb{R}^{H \times W \times d}$ means the output of $j$-th layer of image $\mathbf{x}_i$. Notably, the visual encoder, which is used to extract internal normal features, is modified with V-V attention mechanism to enhance its capability to preserve local information (Li et al., 2024b; Zhou et al., 2024).

**Semantic-Level Alignment** For each training image $\mathbf{x}_i$, the corresponding normal prompt embedding $\mathbf{e}_i^n$ is pulled closer to the normal patch features in $\mathcal{F}_i$, while the anomalous one $\mathbf{e}_i^a$ is pushed away from them. Formally, the semantic-level alignment loss is defined as

$$\mathcal{L}_s^n = \frac{1}{|\mathcal{J}|} \sum_{j \in \mathcal{J}} \left( \frac{1}{H \times W} \sum_{h=1}^{H} \sum_{w=1}^{W} -\log \frac{\exp(<\mathbf{F}_{ij,hw}, \mathbf{e}_i^n>)}{\exp(<\mathbf{F}_{ij,hw}, \mathbf{e}_i^n>) + \exp(<\mathbf{F}_{ij,hw}, \mathbf{e}_i^a>)} \right), \quad (6)$$

where $<\cdot, \cdot>$ denotes cosine similarity while $\mathbf{F}_{ij,hw}$ denotes the feature of $(h, w)$-th patch of image $\mathbf{x}_i$ at the $j$-th layer. By minimizing this loss, the semantics of normal prompts become more aligned with the normal patches, while the semantics of the anomalous prompts diverge from them.

**Token-Level Alignment** Solely aligning the embedding of the entire prompt with patch features may weaken the prompt's ability to capture details, which, however, is crucial for the success of fine-grained anomaly detection tasks. Inspired by the observation that multiple image patches often correspond to a single word in the caption (Bica et al., 2024), we propose to encourage the alignment between a group of patches and the embedding of one prompt token. This approach encourages prompts to capture detailed, highly relevant local information provided by the V-V attention in CLIP. Specifically, we first compute the similarity matrix $\mathbf{M}_{ij} \in \mathbb{R}^{T' \times R}$ between normal prompt token embeddings and patch embeddings from the $j$-th layer of CLIP visual encoder for image $\mathbf{x}_i$, where $T'$ equals to $T + m$ and $R$ equals to $H \times W$. The element $M_{ij,tk}$ of $\mathbf{M}_{ij}$ represents the similarity between prompt token embedding $\mathbf{p}_{i,t}$ and patch feature $\mathbf{v}_{ij,k}$. To obtain the token-level alignment weight, we then normalize the elements of $\mathbf{M}_{ij}$ to the range $[0, 1]$ by min-max normalization across patch dimension as $\hat{M}_{ij,tk} = \frac{M_{ij,tk} - \min_r M_{ij,tr}}{\max_r M_{ij,tr} - \min_r M_{ij,tr}}$. Then we sparsify the normalized matrix to reduce irrelevant patches and encourage each prompt token embedding to be aligned with several patch features by setting $\tilde{M}_{ij,tk} = \hat{M}_{ij,tk}$ if $\hat{M}_{ij,tk} \geq \sigma$ or 0 otherwise, where $\sigma$ denotes the sparsity threshold. The token-level alignment weight is finally obtained as $W_{ij,tk} = \frac{\tilde{M}_{ij,tk}}{\sum_{r=1}^{R} \tilde{M}_{ij,tr}}$. With the token-level alignment weight $\mathbf{W}_{ij}$, we can obtain the representation of patch features that are most relevant to a prompt token as $\tilde{\mathbf{v}}_{ij,t} = \sum_{r=1}^{R} W_{ij,tr} \mathbf{v}_{ij,r}$. That is, the embedding $\tilde{\mathbf{v}}_{ij,t}$ can be understood as the embedding of collective visual patches that are relevant to the $j$-th token. Thus, we propose to pull closer the prompt token embedding $\mathbf{p}_{i,t}$ and the collective visual embedding $\tilde{\mathbf{v}}_{ij,t}$ by minimizing the following loss

$$\mathcal{L}_f^n = \frac{1}{|\mathcal{J}|} \sum_{j \in \mathcal{J}} \left[ -\frac{1}{T'} \sum_{t=1}^{T'} \left( \log \frac{\exp(<\mathbf{p}_{i,t}, \tilde{\mathbf{v}}_{ij,t}>)}{\sum_{k=1}^{T'} \exp(<\mathbf{p}_{i,t}, \tilde{\mathbf{v}}_{ij,k}>)} + \log \frac{\exp(<\tilde{\mathbf{v}}_{ij,t}, \mathbf{p}_{i,t}>)}{\sum_{k=1}^{T'} \exp(<\tilde{\mathbf{v}}_{ij,t}, \mathbf{p}_{i,k}>)} \right) \right]. \quad (7)$$

**Synthetic Visual Guidance for Anomalous Prompt** Due to the lack of anomalous samples, we can only push the anomalous prompt embedding away from normal patch features. However, since anomalies are often visually similar to normal images, blindly pushing the anomalous prompt away can result in the prompt losing meaningful semantic information, rendering it ineffective for anomaly detection. SimpleNet (Liu et al., 2023) proposed to add noise to normal features. Since Gaussian noise is too mild and does not provide any meaningful information to guide the anomalous prompt, we propose to further distort the feature $\mathbf{F}_{ij}$ from $j$-th layer with feature from the $j'$-th layer, where $j'$ is the index immediately before $j$ in $\mathcal{J}$, that is,

$$\mathbf{F}_{ij}^a = \text{Normalize}(\mathbf{F}_{ij} + \mathbf{F}_{ij'} + \epsilon), \quad (8)$$

where $\text{Normalize}(\cdot)$ denotes the operation that normalizes $\mathbf{F}_{ij}^a$ to have a norm of 1, and $\epsilon \sim \mathcal{N}(0, 1)$ denotes the Gaussian noise. By adding features from different layers, information at different scales can be fused, resulting in synthetic visual features $\mathbf{F}_{ij}^a$ that include both low-level visual details and

high-level semantic concepts, thereby introducing some inconsistencies or unnaturalness. Then, we can perform the semantic-level and token-level alignment for the anomalous prompt:

$$\mathcal{L}_s^a = \frac{1}{|\mathcal{J}|} \sum_{j \in \mathcal{J}} \left( \frac{1}{H \times W} \sum_{h=1}^{H} \sum_{w=1}^{W} -\log \frac{\exp(<\mathbf{F}_{ij,hw}^a, \mathbf{e}_i^a>)}{\exp(<\mathbf{F}_{ij,hw}^a, \mathbf{e}_i^a>) + \exp(<\mathbf{F}_{ij,hw}^a, \mathbf{e}_i^n>)} \right), \quad (9)$$

$$\mathcal{L}_f^a = \frac{1}{|\mathcal{J}|} \sum_{j \in \mathcal{J}} \left[ -\frac{1}{T'} \sum_{t=1}^{T'} \left( \log \frac{\exp(<\mathbf{n}_{i,t}, \tilde{\mathbf{v}}_{ij,t}^a>)}{\sum_{k=1}^{T'} \exp(<\mathbf{n}_{i,t}, \tilde{\mathbf{v}}_{ij,k}^a>)} + \log \frac{\exp(<\tilde{\mathbf{v}}_{ij,t}^a, \mathbf{n}_{i,t}>)}{\sum_{k=1}^{T'} \exp(<\tilde{\mathbf{v}}_{ij,t}^a, \mathbf{n}_{i,k}>)} \right) \right], \quad (10)$$

where $\mathbf{n}_{i,t}$ denotes the $t$-th anomalous prompt token embedding and $\tilde{\mathbf{v}}_{ij,t}^a$ denotes the grouping patch features corresponding to $\mathbf{n}_{i,t}$.

### 3.2.2 TEXTUAL GUIDANCE

In the visual guidance above, we can only access normal images and the abnormal visual data are synthetic, which would introduce bias into the prompt learning process. To address this issue, we propose to leverage the manually crafted prompts, which can be understood as a kind of expert knowledge, to train the model. But we follow the one-for-all paradigm and only use general descriptions for normality and abnormality, *e.g.*, replacing any specific category name with the common word 'object'. In particular, we generate normal manually-crafted prompts $\mathcal{T}^n$ from a template list that describes general normality, such as *"A photo of a perfect object, A photo of a flawless object"*, *etc.* While the anomalous manual prompts $\mathcal{T}^a$ are generated from the anomaly labels of the datasets as done in PromptAD, *e.g., "A photo of the object with color stain"*. Completed manually-crafted prompts $\mathcal{T}^n$ and $\mathcal{T}^a$ refer to Appenix A.2. After generating a set of general normality and abnormality descriptions, instead of directly using them for anomaly detection, which lacks specificity and adaptability across different categories, we use them to guide the prompt learning. Consistent with the visual guidance of prompts, contrastive learning is also applied to align the learnable prompts with manual prompts by minimizing the loss

$$\mathcal{L}_t = \mathbb{E}_{\mathbf{t}^n} \left[ -\log \frac{\exp(<\mathbf{e}^n, \mathbf{t}^n>)}{\exp(<\mathbf{e}^n, \mathbf{t}^n>) + \exp(<\mathbf{e}^a, \mathbf{t}^n>)} \right] + \mathbb{E}_{\mathbf{t}^a} \left[ -\log \frac{\exp(<\mathbf{e}^a, \mathbf{t}^a>)}{\exp(<\mathbf{e}^a, \mathbf{t}^a>) + \exp(<\mathbf{e}^n, \mathbf{t}^a>)} \right],$$
$$(11)$$

where $\mathbf{t}^n, \mathbf{t}^a$ denote embeddings of normal manual prompts and anomalous manual prompts, respectively. With the guidance of manual prompts, the learned prompts are endowed with the ability to identify general normality and real abnormality via the use of expert knowledge.

The overall objective of prompt learning is a weighted sum of visual guidance and textual guidance:

$$\mathcal{L} = (\mathcal{L}_s^n + \mathcal{L}_s^a) + \beta(\mathcal{L}_f^n + \mathcal{L}_f^a) + \alpha \mathcal{L}_t \quad (12)$$

where $\alpha$ and $\beta$ are hyperparameters.

### 3.3 CATEGORY-AWARE MEMORY BANK

Due to the rich normality information embedded in the features extracted by CLIP's visual encoder, the memory bank has been introduced in previous works (Jeong et al., 2023; Li et al., 2024b) to enhance anomaly detection. However, under the one-for-all paradigm, the memory bank contains visual features from different classes and anomalous patch features may resemble and have high similarity with normal features from totally different categories, leading to undesired feature matching. This is called a mismatch problem and impairs the effectiveness of the memory bank in distinguishing between normal and anomalous features. To overcome this challenge, a category-aware visual memory bank is proposed. Instead of naively storing the image patch tokens output by the $j$-th layer of the CLIP visual encoder, we make full of category tokens $\mathbf{Z}$ extracted from the Q-Former. Specifically, we store the image patch tokens $\mathbf{F}_{ij}$ of image $\mathbf{x}_i$ and its corresponding Q-Former (*i.e.*, category) tokens $\mathbf{Z}_i$. The category-aware visual memory bank can be represented as

$$\mathbf{B}^v = \{[\mathbf{Z}_1; \mathbf{F}_{1j}], [\mathbf{Z}_2; \mathbf{F}_{2j}], \ldots, [\mathbf{Z}_N; \mathbf{F}_{Nj}]\} . \quad (13)$$

In the testing phase, we first retrieve a group of elements from $\mathbf{B}_v$ using the similarity between the category tokens $\mathbf{Z}$'s of the test image and stored ones. We then evaluate the similarity between patch

token features $\mathbf{F}_{ij}$ of the test image and the ones that are retrieved at the first stage. In practice, the intermediate features of two layers are selected as memory, denoted as $\mathbf{B}_{j_1}^v, \mathbf{B}_{j_2}^v$, where $j_1, j_2 \in \mathcal{J}$. Besides constructing a visual memory bank, we also build the category-aware prompt embedding bank $\mathbf{B}^p$ which stores various prompt embeddings of training instances from $\mathcal{X}_{train}$ as

$$\mathbf{B}^p = \{[\mathbf{Z}_1; \mathbf{e}_1^n; \mathbf{e}_1^a], [\mathbf{Z}_2; \mathbf{e}_2^n; \mathbf{e}_2^a], \dots, [\mathbf{Z}_N; \mathbf{e}_N^n; \mathbf{e}_N^a]\}. \tag{14}$$

During testing, we retrieve the top-$k$ prompts most similar to the test image, following a strategy similar to that used in the category-aware visual memory bank.

## 3.4 Anomaly Detection

Unlike prior works that rely solely on fixed prompts learned during the training phase (Zhou et al., 2024; Li et al., 2024b), we can additionally leverage information from the test image by our prompt generator. With the groups of prompt embeddings $\mathcal{E}^n$ and $\mathcal{E}^a$ retrieved from $\mathbf{B}^p$ as stated above, the prompt embeddings $\mathbf{e}_t^n$ and $\mathbf{e}_t^a$ of the test image for detecting anomalies are then computed as

$$\bar{\mathbf{e}}^n = \frac{1}{2}\left(\frac{1}{K}\sum_{\mathbf{e}\in\mathcal{E}^n}\mathbf{e} + \mathbf{e}_t^n\right), \qquad \bar{\mathbf{e}}^a = \frac{1}{2}\left(\frac{1}{K}\sum_{\mathbf{e}\in\mathcal{E}^a}\mathbf{e} + \mathbf{e}_t^a\right), \tag{15}$$

where $\mathbf{e}_t^n$ and $\mathbf{e}_t^a$ denote normal and anomalous fusion prompt embeddings of the test image, respectively. Given the $j$-th layer patch tokens $\mathbf{F}_{\cdot j} \in \mathbb{R}^{H \times W \times d}$ of test image with $\cdot$ representing the test image, the $(h, w)$-th element of prompt-guided anomaly map $\mathbf{A}_j^p \in [0, 1]^{H \times W}$ is obtained as

$$\mathbf{A}_{j,hw}^p = \frac{\exp(<\mathbf{F}_{\cdot j,hw}, \bar{\mathbf{e}}^a>)}{\exp(<\mathbf{F}_{\cdot j,hw}, \bar{\mathbf{e}}^a>) + \exp(<\mathbf{F}_{\cdot j,hw}, \bar{\mathbf{e}}^n>)}. \tag{16}$$

As there exist selected features from $\mathcal{J}$, the final prompt-guided anomaly map is calculated as the average of that in all $|\mathcal{J}|$ layers, that is, $\bar{\mathbf{A}}^p = \frac{1}{|\mathcal{J}|}\sum_{j\in\mathcal{J}}\mathbf{A}_j^p$. In addition, we also make use of the category-aware visual memory bank. With the group of patch tokens $\mathcal{V}_1, \mathcal{V}_2$ retrieved from $\mathbf{B}_{j_1}^v, \mathbf{B}_{j_2}^v$, the layer patch tokens $\mathbf{F}_{\cdot j_1}, \mathbf{F}_{\cdot j_2}$ of test image is compared with $\mathcal{V}_1, \mathcal{V}_2$ to obtain visual anomaly map $\mathbf{A}^v \in [0, 1]^{H \times W}$ as follows:

$$\mathbf{A}_{hw}^v = \min_{\mathbf{v}\in\mathcal{V}_1}\frac{1}{2}(1 - <\mathbf{F}_{\cdot j_1,hw}, \mathbf{v}>) + \min_{\mathbf{v}\in\mathcal{V}_2}\frac{1}{2}(1 - <\mathbf{F}_{\cdot j_2,hw}, \mathbf{v}>), \tag{17}$$

where $j_1$ and $j_2$ denote the two layer indices that are used in construction of $\mathbf{B}_{j_1}^v$ and $\mathbf{B}_{j_2}^v$, respectively. The final pixel-level anomaly score map and image-level score are computed as

$$\mathbf{A}_{pixel} = \bar{\mathbf{A}}^p + \mathbf{A}^v, \qquad \mathbf{A}_{image} = \max_{h,w}\bar{\mathbf{A}}^p + \max_{h,w}\mathbf{A}^v, \tag{18}$$

where the image-level score is sum of the maxima of the prompt-guided and visual anomaly maps.

## 4 Experiments

### 4.1 Experiment Setting and Baseline

**Dataset** In this paper, we conduct extensive experiments on benchmarks MVTec (Bergmann et al., 2019) and VisA (Zou et al., 2022). The MvTec dataset contains 10 object categories and 5 texture categories while VisA dataset contains 12 different categories. Both benchmarks comprise high-resolution images and various anomalous types.

**Experiment Setup** We conduct the comparison experiments between our method and the latest methods under the setting that the training dataset contains all categories of the benchmark and only $K$-shot normal image for each class. Additionally, the memory bank used across all methods, as well as the prompts in CLIP-based methods, follow the one-for-all paradigm. This means that the memory bank contains features from various categories, and the prompts are not initialized or specifically tailored to each individual class. The experimental results reported in this section are averaged across their respective sub-datasets.

**Evaluation** Referring to previous work, the performance of anomaly detection and localization is evaluated using the Area Under the Receiver Operating Characteristic Curve (AUROC). Additionally, the Area Under Precision Recall (AUPR) for anomaly detection and the Per-Region Overlap

Table 1: Comparison of anomaly detection and localization performance on MVTec and VisA datasets across different few-shot settings under the one-for-all paradigm. The best result is marked in bold and the runner-up result is marked underlined.

| Setup | Method | MVTec | | | | VisA | | | |
|---|---|---|---|---|---|---|---|---|---|
| | | Image-level | | Pixel-level | | Image-level | | Pixel-level | |
| | | AUROC | AUPR | AUROC | PRO | AUROC | AUPR | AUROC | PRO |
| 0-shot | WinCLIP | 89.1 | 95.0 | 81.1 | 62.3 | 73.3 | 76.3 | 79.8 | 57.9 |
| | AnomalyCLIP | 91.5 | 96.2 | 91.1 | 81.4 | 82.1 | 85.4 | 95.5 | 87.0 |
| 1-shot | SPADE | 58.8 | 63.7 | 60.4 | 53.1 | 61.3 | 68.2 | 69.0 | 57.2 |
| | PatchCore | 63.7 | 81.2 | 83.9 | 72.7 | 58.9 | 62.8 | 76.7 | 64.3 |
| | FastRecon | 51.2 | 72.6 | 62.1 | 60.3 | 55.0 | 72.8 | 70.7 | 58.2 |
| | WinCLIP | 92.8 | 96.5 | 92.4 | 83.5 | 83.1 | 85.1 | 94.6 | 80.9 |
| | PromptAD | 86.3 | 93.4 | 91.8 | 83.6 | 80.8 | 83.2 | 96.3 | 82.2 |
| | InCTRL | - | - | - | - | - | - | - | - |
| | IIPAD | 94.2 | 97.2 | 96.4 | 89.8 | 85.4 | 87.5 | 96.9 | 87.3 |
| 2-shot | SPADE | 68.4 | 84.2 | 61.2 | 54.7 | 66.8 | 72.0 | 71.3 | 59.6 |
| | PatchCore | 72.4 | 86.2 | 89.6 | 74.2 | 60.2 | 64.3 | 82.4 | 68.1 |
| | FastRecon | 51.7 | 74.9 | 62.4 | 59.9 | 58.2 | 74.6 | 79.6 | 63.5 |
| | WinCLIP | 92.7 | 96.3 | 92.4 | 83.9 | 83.7 | 84.9 | 95.1 | 81.8 |
| | PromptAD | 89.2 | 94.8 | 92.2 | 84.3 | 84.3 | 87.8 | 96.9 | 84.7 |
| | InCTRL | 94.0 | 96.9 | - | - | 85.8 | 87.7 | - | - |
| | IIPAD | 95.7 | 97.9 | 96.7 | 90.3 | 86.7 | 88.6 | 97.2 | 87.9 |
| 4-shot | SPADE | 76.6 | 88.8 | 62.8 | 55.6 | 73.0 | 76.6 | 72.1 | 60.9 |
| | PatchCore | 74.9 | 88.8 | 92.6 | 80.8 | 62.6 | 69.9 | 85.4 | 70.6 |
| | FastRecon | 50.8 | 73.1 | 65.0 | 62.8 | 57.6 | 73.7 | 78.8 | 62.9 |
| | WinCLIP | 94.0 | 96.9 | 92.9 | 84.4 | 84.1 | 86.1 | 95.2 | 82.1 |
| | PromptAD | 90.6 | 96.5 | 92.4 | 84.6 | 85.7 | 88.8 | 97.2 | 84.7 |
| | InCTRL | 94.5 | 97.2 | - | - | 87.7 | 90.2 | - | - |
| | IIPAD | 96.1 | 98.1 | 97.0 | 91.2 | 88.3 | 89.6 | 97.4 | 88.3 |

(PRO) for anomaly localization are employed to provide a more comprehensive analysis of the model's performance.

**Implementation details** We build our model based on publicly available CLIP (Ilharco et al., 2021) (ViT-L/14). The Q-Former used to extract category tokens is taken from BLIP-Diffusion (Li et al., 2024a). All parameters of pre-trained models are frozen. The length of learnable prompt tokens is set as 24. All experiments are conducted with a single NVIDIA RTX 3090 24GB GPU. For more implementation details refer to Appendix A.1.

**Baselines** To demonstrate the superiority of our method, we select the lateset training-free or CLIP-based anomaly detection methods as baselines: SPADE (Cohen & Hoshen, 2020), PatchCore (Roth et al., 2022), FastRecon (Fang et al., 2023), WinCLIP (Jeong et al., 2023), PromptAD (Li et al., 2024b), InCTRL (Zhu & Pang, 2024) and AnomalyCLIP (Zhou et al., 2024). Then, we further compare with the full-shot one-for-all anomaly detection methods: UniAD (You et al., 2022), OmniAL (Zhao, 2023), HVQ-Trans (Lu et al., 2023) and DiAD (He et al., 2024). For more baselines details refer to Appendix A.3.

## 4.2 EXPERIMENT RESULTS

**Overall Performance** The experiment results on MVTec and VisA datasets in different few-shot setting are demonstrated in Table 1, where SPADE (Cohen & Hoshen, 2020), PatchCore (Roth et al., 2022) are reformulated in the few-shot setting. To further demonstrate the competitiveness of our method, as shown in Table 2, we also compare our method, which remains the one-for-all paradigm, with baselines in the 'one-for-one' paradigm and the results of them are referred to their original papers. As shown in Table 1- 2, the performance of zero-shot methods remains weak compared to other baselines, even though they utilize auxiliary datasets or additional expert knowledge. This indicates the importance of incorporating information from the target domain for effective anomaly detection. Comparing Table 1 and Table 2, it is evident that the performance of all baselines significantly drops when they are trained or adapted to the one-for-all paradigm. This indicates that these few-shot anomaly detection methods struggle to effectively detect anomalies across different categories when using a unified model. In contrast, our method, which trains an instance-specific prompt generator, significantly outperforms other baselines in both pixel-level and image-level anomaly detection under the one-for-all paradigm. Notably, even in the 'one-for-one' paradigm, our method surpasses all baselines at the pixel level and achieves the second-best performance at the image level

Table 2: Comparison of anomaly detection and localization performance on MVTec and VisA across different few-shot settings under the one-for-one paradigm. The results of baselines are taken from the original paper. Our method remains the one-for-all paradigm. The best result is marked in bold and the runner-up result is marked underlined.

| Setup | Method | MVTec | | | | VisA | | | |
|---|---|---|---|---|---|---|---|---|---|
| | | Image-level | | Pixel-level | | Image-level | | Pixel-level | |
| | | AUROC | AUPR | AUROC | PRO | AUROC | AUPR | AUROC | PRO |
| 0-shot | WinCLIP | 91.8 | 96.5 | 85.1 | 64.6 | 78.1 | 81.2 | 79.6 | 56.8 |
| 1-shot | SPADE | 81.0 | 90.6 | 91.2 | 83.9 | 79.5 | 82.0 | 95.6 | 84.1 |
| | PatchCore | 83.4 | 92.2 | 92.0 | 79.7 | 79.9 | 82.8 | 95.4 | 80.5 |
| | FastRecon | - | - | - | - | - | - | - | - |
| | WinCLIP | 93.1 | 96.5 | 95.2 | 87.1 | 83.8 | 85.1 | 96.4 | 85.1 |
| | PromptAD | **94.6** | 97.1 | 95.9 | 87.9 | **86.9** | **88.4** | 96.7 | 85.1 |
| | IIPAD(One-for-All) | 94.2 | **97.2** | **96.4** | **89.8** | 85.4 | 87.5 | **96.9** | **87.3** |
| 2-shot | SPADE | 82.9 | 91.7 | 92.0 | 85.7 | 80.7 | 82.3 | 96.2 | 85.7 |
| | PatchCore | 86.3 | 93.8 | 93.3 | 82.3 | 81.6 | 84.8 | 96.1 | 82.6 |
| | FastRecon | 91.0 | - | 95.9 | - | - | - | - | - |
| | WinCLIP | 94.4 | 97.0 | 96.0 | 88.4 | 84.6 | 85.8 | 96.8 | 86.2 |
| | PromptAD | **95.7** | **97.9** | 96.2 | 88.5 | **88.3** | **90.0** | 97.1 | 85.8 |
| | IIPAD(One-for-All) | **95.7** | **97.9** | **96.7** | **90.3** | 86.7 | 88.6 | **97.2** | **87.9** |
| 4-shot | SPADE | 84.8 | 92.5 | 92.7 | 87.0 | 81.7 | 83.4 | 96.6 | 87.3 |
| | PatchCore | 88.8 | 94.5 | 94.3 | 84.3 | 85.3 | 87.5 | 96.8 | 84.9 |
| | FastRecon | 94.2 | - | **97.0** | - | - | - | - | - |
| | WinCLIP | 95.2 | 97.3 | 96.2 | 89.0 | 87.3 | 88.8 | 97.2 | 87.6 |
| | PromptAD | **96.6** | **98.5** | 96.5 | 90.5 | **89.1** | **90.8** | **97.4** | 86.2 |
| | IIPAD(One-for-All) | 96.1 | 98.1 | **97.0** | **91.2** | 88.3 | 89.6 | **97.4** | **88.3** |

Figure 2: Visualization result comparison of 1-shot anomaly detection on MVTec and VisA.

across various few-shot settings on the MVTec and VisA datasets, further demonstrating superiority of our method. More comparative analysis of our method refer to Appendix A.4

**Comparison with One-For-All Full-Shot Methods** The results, presented in Table 3, show that our method achieves performance close to state-of-the-art full-shot one-for-all anomaly detection approaches, despite using only a few normal images. Remarkably, with the 4-shot setting, our method outperforms UniAD and DiAD at the pixel level on the MVTec dataset and surpasses OmniAL and DiAD at both image and pixel levels on the VisA dataset. This demonstrates that our method can effectively leverage limited data.

Table 3: Comparison of anomaly detection and localization performance on MVTec and VisA with full-shot one-for-all anomaly detection methods.

| Method | Setup | MVTec | | VisA | |
|---|---|---|---|---|---|
| | | $AUROC_I$ | $AUROC_P$ | $AUROC_I$ | $AUROC_P$ |
| IIPAD | 1-shot | 94.2 | 96.4 | 85.4 | 96.9 |
| | 2-shot | 95.6 | 96.7 | 86.7 | 97.2 |
| | 4-shot | 96.1 | 97.0 | 88.3 | 97.4 |
| UniAD | | 96.5 | 96.8 | 91.9 | 98.6 |
| OmniAL | Full-shot | 97.2 | 98.3 | 87.8 | 96.6 |
| DiAD | | 97.2 | 96.8 | 86.8 | 96.0 |
| HVQ-Trans | | 98.0 | 97.3 | 93.2 | 98.7 |

**Visualization Results** We conducted qualitative experiments on MVTec and VisA. The partial visualization results are shown in Figure 2. It can be observed that our method provides more distinguishing and precise anomaly maps compared to other methods. Notably, as shown in Figure 2, our method is more sensitive to anomaly compared to PromptAD while avoiding misclassification of normal regions like WinCLIP does. Additionally, our method shows the efficiency of localizing small anomalies like those occur in candles and capsules. For more qualitative results refer to A.7.

### 4.3 ABLATION STUDY

**Prompt Generator Ablation** To validate the effectiveness of our prompt generator design, we conducted an ablation study by removing $M_1$ and $M_2$ components of generated prompts. $M_1$ denotes the instance-specific visual tokens $\mathbf{C}^n, \mathbf{C}^a$, while $M_2$ denotes the normal and anomalous object tokens $\mathbf{O^n}, \mathbf{O^a}$ appended at the end of prompts. As demonstrated in Table 4, the performance of our method declines when $M_2$ is removed, indicating that detailed and projected category descriptions are

Table 4: Prompt generator ablation on MVTec and VisA under 1-shot setting.

| $M_1$ | $M_2$ | MVTec | | VisA | |
|---|---|---|---|---|---|
| | | $\text{AUROC}_I$ | $\text{AUROC}_P$ | $\text{AUROC}_I$ | $\text{AUROC}_P$ |
| × | × | 90.6 | 95.8 | 83.7 | 96.2 |
| ✓ | × | 93.2 | 96.2 | 83.8 | 96.7 |
| × | ✓ | 92.5 | 96.2 | 84.1 | 96.4 |
| ✓ | ✓ | **94.2** | **96.4** | **85.3** | **96.9** |

beneficial for anomaly detection. Additionally, the introduction of instance-specific component, which incorporate richer image information to prompts, improves the prompt's ability to detect and localize anomalies. More ablation and detailed experiment results refer to Appendix A.5.

**Loss Ablation** To study the effectiveness of each guidance used in our method, we conducted a loss ablation study by sequentially removing different loss components to assess their individual contributions. As illustrated in the Table 5, where $\mathcal{L}_s$, $\mathcal{L}_t$ and $\mathcal{L}_f$ refer to the semantic alignment loss, textual guidance loss and token-level alignment loss respectively, both image-level and pixel-level performances of our method consistently degrade as we remove these loss components. A significant performance drop can be observed when

Table 5: Loss ablation on MVTec and VisA under 1-shot setting.

| $\mathcal{L}_s$ | $\mathcal{L}_t$ | $\mathcal{L}_f$ | MVTec | | VisA | |
|---|---|---|---|---|---|---|
| | | | $\text{AUROC}_I$ | $\text{AUROC}_P$ | $\text{AUROC}_I$ | $\text{AUROC}_P$ |
| ✓ | × | × | 91.1 | 95.8 | 81.5 | 95.9 |
| ✓ | × | ✓ | 92.3 | 96.0 | 81.8 | 96.2 |
| ✓ | ✓ | × | 91.4 | 96.2 | 82.6 | 96.4 |
| × | ✓ | ✓ | 92.6 | 95.6 | 83.1 | 96.2 |
| ✓ | ✓ | ✓ | **94.2** | **96.4** | **85.4** | **96.9** |

the semantic-level alignment is removed. This is because semantic-level alignment serves as the primary objective for training the prompt generator, aligning with the detection behavior during testing. The token-level alignment loss encourages the prompt tokens to focus on specific patches, enhancing the anomaly localization capability of the generated prompts. Moreover, the improvement in performance brought by text guidance is significant. This indicates that expert knowledge, even when represented in text form rather than as specific anomaly maps, is very helpful for anomaly detection. These losses cooperatively guide the prompt generator to produce prompts that capture the normality and abnormality of images.

**Category-Aware Memory Bank** To study the effectiveness of our category-aware memory bank design, we conducted an ablation study by either removing or degenerating the category-aware memory bank. The results are presented in Table 6, where OVB denotes using the original visual bank without utilizing the Q-Former tokens $\mathbf{Z}_i$ and CAVB denotes using the proposed category-aware visual bank. The first row shows

Table 6: Visual memory bank ablation on MVTec and VisA under 1-shot setting.

| OVB | CAVB | MVTec | | VisA | |
|---|---|---|---|---|---|
| | | $\text{AUROC}_I$ | $\text{AUROC}_P$ | $\text{AUROC}_I$ | $\text{AUROC}_P$ |
| × | × | 91.7 | 89.5 | 82.3 | 93.6 |
| ✓ | × | 93.1 | 95.9 | 84.5 | 96.7 |
| × | ✓ | **94.2** | **96.4** | **85.4** | **96.9** |

the performance of the prompt generator in isolation from visual memory bank. A significant performance drop occurs when the visual memory bank is removed, indicating that the visual memory bank plays a critical role in our proposed one-for-all anomaly detection method. Notably, the category-aware visual memory bank outperforms the original version, indicating that it can accurately retrieve the corresponding sub-bank based on Q-Former token similarity, effectively addressing the mismatch issues present in the original visual memory bank.

## 5 CONCLUSION

In this paper, we propose a novel few-shot anomaly detection method that enables to detect anomalies across different categories with a unified model. We design a class-shared prompt generator, which can adaptively generate instance-specific prompts for detecting anomalies across different classes. Then, the prompt generator is trained by aligning the prompt with both visual and textual feature guidance. Furthermore, we address the mismatch issue in the memory bank to adopt it in one-for-all paradigm. The incorporation of test-time knowledge further enhances the adaptability and flexibility of the prompts. Extensive experimental results on MVTec and VisA demonstrate the superiority of our method in the few-normal-shot setting within the one-for-all paradigm.

## ACKNOWLEDGEMENTS

This work is supported by the National Natural Science Foundation of China (No. 62276280, 62276279), Guangzhou Science and Technology Planning Project (No. 2024A04J9967).

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

## A  APPENDIX

### A.1  IMPLEMENT DETAILS

**Training Details**  We build our model based on publicly available CLIP (`VIT-L-14`). The Q-Former used to extract category tokens is taken from BLIP-Diffusion, which is already align to CLIP's text encoder. The parameters of CLIP and Q-Former are all frozen during training. The length of learnable prompt $T$ is set as 24. The length of compressed category tokens $m$ is 4. The projection networks used to incorporate visual information are two-layer neural network. The weight factors $\alpha, \beta$ that balance loss are both fixed as 8. To provide adequate visual details, we select patch visual embeddings $\mathbf{F}_{ij}$ from the 6-th, 12-th, 18-th, and 24-th layers of the visual encoder to guide prompt learning. The visual encoder is reformed into V-V attention mechanism. Patch features from the 6-th and 18-th layers of the original visual encoder are selected as memory bank. The threshold for sparsify the normalized matrix is set as $\frac{1}{256}$. We use the Adam optimizer (Kingma & Ba, 2015) with a learning rate of 0.001 to update model parameters. The epoch is 20 for all experiments, which are performed in PyTorch with a single NVIDIA RTX 3090 24GB GPU.

### A.2  MANUAL PROMPTS DETAILS

We introduce manual text prompt as expert knowledge to guide learnable prompts in capturing normality and abnormality. The manual normal and anomalous prompts are given as follows:

- Normal manual prompts: {`a image of the normal object`, `a image of the flawless object`, `a image of the perfect object`, `a image of the unblemished object`, `a image of the object without flaw`, `a image of the object without defect`, `a image of the object without damage`}

- Anomalous manual prompts (MVTec): { `a image of the damaged object`, `a image of the abnormal object`, `a image of the imperfect object`, `a image of the blemished object`, `a image of the object with flaw`, `a image of the object with defect`, `a image of the object with damage`, `a image of the object with large breakage`, `a image of the object with small breakage`, `a image of the object with contamination`, `a image of the object with defect`, `a image of the object with anomaly`, `a image of the object with hole`, `a image of the object with color stain`, `a image of the object with metal contamination`, `a image of the object with thread residue`, `a image of the object with thread`, `a image of the object with cut`, `a image of the object with crack`, `a image of the object with cut`, `a image of the object with hole`, `a image of the object with print`, `a image of the object with color stain`, `a image of the object with cut`, `a image of the object with fold`, `a image of the object with glue`, `a image of the object with poke`, `a image of the object with bent wire`, `a image of the object with missing part`, `a image of the object with missing wire`, `a image of the object with cut`, `a image of the object with poke`, `a image of the object with crack`, `a image of the object with faulty imprint`, `a image of the object with poke`, `a image of the object with scratch`, `a image of the object squeezed with compression`, `a image of the object with breakage`, `a image of the object with thread residue`, `a image of the object with thread`, `a image of the object with metal contamination` }

- Anomalous manual prompts (VisA): { `a image of the damaged object`, `a image of the abnormal object`, `a image of the imperfect object`, `a image of the blemished object`, `a image of the`

```
object with flaw', 'a image of the object with defect', 'a
image of the object with damage', 'a image of the object
with melded wax', 'a image of the object with foreign
particals', 'a image of the object with extra wax', 'a
image of the object with chunk of wax missing', 'a image of
the object with weird candle wick', 'a image of the object
with damaged corner of packaging', 'a image of the object
with different colour spot', 'a image of the object with
scratch', 'a image of the object with discolor', 'a image
of the object with misshape', 'a image of the object with
leak', 'a image of the object with bubble', 'a image of the
object with breakage', 'a image of the object with small
scratches', 'a image of the object with burnt', 'a image of
the object with stuck together', 'a image of the object with
spot', 'a image of the object with corner missing', 'a image
of the object with scratches', 'a image of the object with
chunk of gum missing', 'a image of the object with colour
spot', 'a image of the object with cracks', 'a image of
the object with color spot', 'a image of the object with
fryum stuck together', 'a image of the object with small
chip around edge', 'a image of the object with bent', 'a
image of the object with missing', 'a image of the object
with melt', 'a image of the object with extra', 'a image of
the object with wrong place', 'a image of the object with
damage', 'a image of the object with dirt' }
```

## A.3 BASELINES DETAILS

We compare our method against a wide range of baselines to demonstrate its superiority. The implementation details of the baselines are provided below:

- SPADE (Cohen & Hoshen, 2020). SPADE uses correspondences based on a multi-resolution feature pyramid, which is inspired by KNN and training-free. SPADE is suitable for detecting anomalies in few-normal-shot setting.

- PatchCore (Roth et al., 2022). PatchCore is also a training-free anomaly detection method which is proposed to construct coreset of normal feature. PatchCore has demonstrate the ability that handle few-shot anomaly detection anomaly detection.

- FastRecon (Fang et al., 2023). FastRecon is proposed to improve the performance of Patch-Core in few-shot settings. This method learns a transform matrix from a few normal samples to reconstruct feature as normal. FastRecon is reproduced from open source code. All parameters are kept the same as in their paper.

- WinCLIP (Jeong et al., 2023). WinCLIP is pioneering work that adopts CLIP in anomaly detection. It utilize manual text prompts to detect anomalies across predefined multi-scale windows. Simultaneously, the multi-scale memory bank is constructed for feature matching in few-shot setting. WinCLIP is reproduced from open source code. All parameters are kept the same as in their paper.

- AnomalyCLIP (Zhou et al., 2024). AnomalyCLIP is the first work to take category information into consideration. AnomalyCLIP is proposed to learn the category-agnostic prompts to capture general normality and abnormality by leveraging auxiliary dataset. AnomalyCLIP is reproduced from its official open source code. All parameters are kept the same as in their paper.

- PromptAD (Li et al., 2024b). PromptAD is a SOTA few-shot anomaly detection method. They proposed semantic concatenation to reverse prompt's semantic and directly optimize a set of learnable context vectors. PromptAD is reproduced from its official open source code. All parameters are kept the same as in their paper.

- InCTRL (Li et al., 2024b). InCTRL is a SOTA few-shot anomaly detection method. They propose to learn the ability to compare on an auxiliary dataset through in-context residual learning to detect anomalies. The performance of InCTRL is referred to its original paper.

## A.4    COMPARISON ANALYSIS

Our method have superiority in the scenarios which requires the anomaly detection methods follow the one-for-all paradigm. For example, we may only access to the raw dataset without category information of it in practice. In this scenario, we can not know the exact category the image belong to, which prohibit WinCLIP and PromptAD to construct different prompt for different category. Our method, on the other hand, follows the one-for-all paradigm and extracts category information directly from the image using Q-Former, thus avoiding the limitations faced by WinCLIP and PromptAD. The superiority of our method in such scenario can be demonstrate by the comparison with other baselines in one-for-all paradigm as shown in Table 1.

## A.5    ADDITIONAL RESULTS AND ABLATIONS

### A.5.1    SYNTHETIC VISUAL FEATURES

To further investigate the impact of synthetic visual guidance, we conducted an ablation study on the synthetic visual feature. As shown in the Table 7, the performance of our method with synthetic visual feature outperforms the version without them. This can be attributed to the synthetic visual feature acting as an anchor with informative context for anomalous prompts to pull closer, rather than blindly pushing away from normal features.

### A.5.2    NEURAL NETWORKS IN PROMPT GENERATOR

We further investigate the impact of modules in instance-specific prompt generator. As shown in the Table 8, every module in the prompt generator has a positive influence. The Projection Network projects the token from the Q-Former and the class token from the VIT into representations that are more suitable for the anomaly detection task. The Cross-Attention module demonstrates an improvement in anomaly localization, indicating that the interaction between the Q-Former and VIT enhances the fine-grained description of prompts.

### A.5.3    GAUSSIAN NOISE IN SYNTHETIC VISUAL FEATURES

To verify whether the Gaussian noise added in synthetic visual features is required or not, we conduct the ablation study and the experiment result is shown in Table 9. As we can observe, a significant performance drop occurs when removing the Gaussian noise in synthetic visual feature. This can be attributed to the role of Gaussian noise in smoothing the synthetic visual feature and pushing it into low-density areas. Consequently, the Gaussian noise is required and plays an important role in synthetic visual feature.

### A.5.4    INTRODUCE MANUAL PROMPT ABLATION

We further investigate the impact of manual prompts in guiding prompt learning by conducting an ablation study comparing the performance of prompts guided by specific manual text descriptions (e.g., $\mathcal{T}^a$) versus those guided by more general manual text descriptions. The results, shown in Table 10, reveal that prompts guided by specific manual text descriptions outperform those guided by general descriptions. This is because specific manual text guidance incorporates knowledge of real anomalies, thereby enhancing the prompts' ability to effectively detect anomalies.

### A.5.5    MULTI-LAYER PATCH FEATURE ABLATION

In the visual guidance phase, we use a selected feature set $\mathcal{F}_i$. In particular, patch visual embeddings $\mathbf{F}_{ij}$ from the 6-th, 12-th, 18-th, and 24-th layers of the visual encoder are selected to guide prompt learning. We conduct the ablation experiment to study the performance of prompt guided by multi-layer features and guided by last layer output. The ablation result is presented in Table 11. It can be observed that prompts guided by multi-layer visual features outperform prompts guided by last

Table 7: Synthetic visual features ablation.

| Synthetic Visual Features | MVTec $AUROC_I$ | MVTec $AUROC_P$ | VisA $AUROC_I$ | VisA $AUROC_P$ |
|:---:|:---:|:---:|:---:|:---:|
| ✓ | 93.6 | 95.9 | 84.3 | 96.5 |
| ✗ | 94.2 | 96.4 | 85.4 | 96.9 |

Table 8: Neural networks in prompt generator ablation.

| Projection Network Relevant to $\mathbf{C}^n$ and $\mathbf{C}^a$ | Cross Attention | Projection Network Relevant to $\mathbf{O}^n$ and $\mathbf{O}^a$ | MVTec $AUROC_I$ | MVTec $AUROC_P$ | VisA $AUROC_I$ | VisA $AUROC_P$ |
|:---:|:---:|:---:|:---:|:---:|:---:|:---:|
| ✗ | ✗ | ✓ | 93.7 | 95.9 | 84.4 | 96.3 |
| ✗ | ✗ | ✓ | 93.7 | 96.1 | 84.6 | 96.5 |
| ✓ | ✗ | ✓ | 94.1 | 96.1 | 84.8 | 96.6 |
| ✗ | ✓ | ✓ | 93.8 | 96.2 | 84.4 | 96.7 |
| ✓ | ✓ | ✓ | 94.2 | 96.4 | 85.4 | 96.9 |

layer output. This indicates that multi-layer visual features contain richer visual information range from low-level details to high level semantic, which can better guide prompts in capturing normality and abnormality.

### A.6   Hyper-parameter Analysis

We conduct hyper-parameter analysis experiments on $\alpha$, $\beta$, and T, where $\alpha$ and $\beta$ represent the weights of the text guidance loss and fine-grained visual guidance loss, respectively, and T is the number of learnable prompt tokens. The experimental results are shown in Table 12. From Table 12a, $\alpha$ has significant influence to our method. Setting $\alpha$ too small results in the prompts capturing insufficient expert knowledge, while setting it too large leads to a loss of visual information. Our method is not sensitive to $\beta$ as shown in Table 12b. For the number of learnable prompt tokens, we find it too few tokens fail to capture the normality and abnormality while too many tokens cause semantic redundancy, so we set T as 24 to achieve the best performance as shown in Table 12c.

### A.6.1   Class-Wise Results

In this section, we report the detailed subset-wise result of our method. The results of MVTec are presented in Table 13- 16. The results of VisA are presented in Table 17- 20.

### A.7   More Visualization Results

### A.7.1   Visualization of Anomaly Localization

Additional qualitative results, which are tested on MVTec and VisA and obtained from our method in 1-shot setting, is shown in Figure 4. From Figure 4, we can observed that our method can detect and locate both large anomaly area and small scratch or dot on the surface. Moreover, our method is very restrained in locating anomalies to avoid identifying normal areas as anomalies.

### A.7.2   Failure Cases Study

In this section, we analysis the failure cares of anomaly localization in MVTec and VisA dataset. In the "Capsules" case, bubbles within the capsules are annotated as anomalies. However, the presence of an uncommon shadow in the image led our method to mistakenly identify this area, resulting in mislocalization. In logical anomaly cases such as the misplacement of a transistor or the sticking of two cashews, as well as in the "Pill" case, the anomalous areas exhibit patterns similar to normal ones. Since we can only access to normal sample and the anomalous labels do not specifically describe these abnormalities, our method struggles to learn the distinguishing features of the anomalies from the text prompts based on such labels, leading to the failure in accurately locating these anomalous areas. The "Zipper" case in MVTec and the "PCB1" case in VisA, may be caused by the

Table 9: Gaussian noise in synthetic visual features ablation.

| Gaussian Noise | MVTec $AUROC_I$ | MVTec $AUROC_P$ | VisA $AUROC_I$ | VisA $AUROC_P$ |
|---|---|---|---|---|
| $\times$ | 92.5 | 95.4 | 83.1 | 96.4 |
| $\checkmark$ | 94.2 | 96.4 | 85.4 | 96.9 |

Table 10: Manual prompt design ablation.

| Generic Prompt | Designed Prompt | MVTec $AUROC_I$ | MVTec $AUROC_P$ | VisA $AUROC_I$ | VisA $AUROC_P$ |
|---|---|---|---|---|---|
| $\checkmark$ | $\times$ | 93.9 | 96.1 | 85.1 | 96.4 |
| $\times$ | $\checkmark$ | 94.2 | 96.4 | 85.4 | 96.9 |

Q-Former's failure to effectively capture and describe the foreground object, leading to an inability to detect small anomalies present on the object.

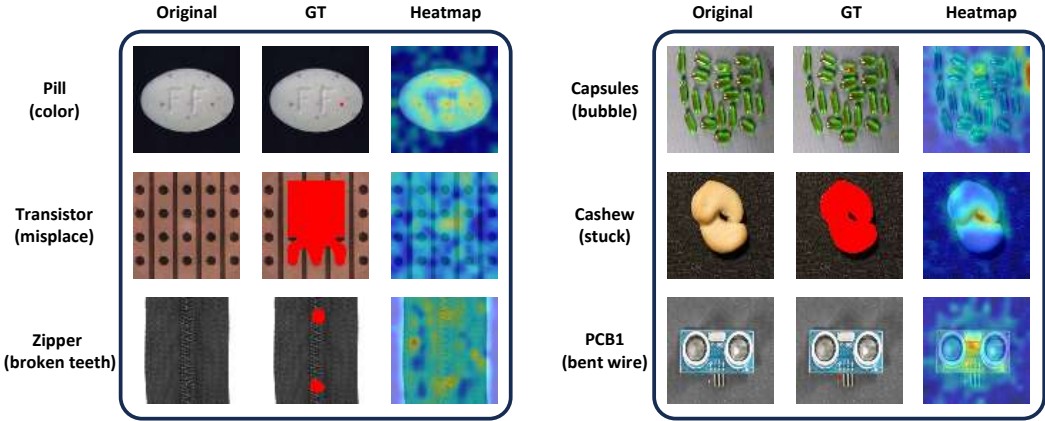

Figure 3: Failure case on MVTec-AD and VisA. The left figure demonstrate the sample from MVTec while the right figure demonstrate the sample from VisA.

Table 11: Layer selection of visual guidance ablation.

| Last Layer | Multi-layer | MVTec AUROC$_I$ | MVTec AUROC$_P$ | VisA AUROC$_I$ | VisA AUROC$_P$ |
|---|---|---|---|---|---|
| ✓ | × | 92.3 | 96.2 | 85.3 | 96.7 |
| × | ✓ | 94.2 | 96.4 | 85.4 | 96.9 |

Table 12: Hyper-parameter analysis result.

(a) Analysis of $\alpha$.

| $\alpha$ | Mvtec Image | Mvtec Pixel | VisA Image | VisA Pixel |
|---|---|---|---|---|
| 1 | 92.1 | 96.0 | 79.9 | 96.1 |
| 4 | 93.5 | 96.3 | 82.4 | 96.4 |
| 8 | **94.2** | **96.4** | **85.4** | **96.9** |
| 12 | 91.4 | 96.2 | 84.5 | 96.8 |

(b) Analysis of $\beta$.

| $\beta$ | Mvtec Image | Mvtec Pixel | VisA Image | VisA Pixel |
|---|---|---|---|---|
| 1 | 93.1 | 96.3 | 84.1 | 96.7 |
| 4 | 93.7 | 96.3 | 84.3 | 96.6 |
| 8 | **94.2** | **96.4** | **85.4** | **96.9** |
| 12 | 93.8 | 96.4 | 84.7 | 96.8 |

(c) Analysis of token number.

| T | Mvtec Image | Mvtec Pixel | VisA Image | VisA Pixel |
|---|---|---|---|---|
| 8 | 93.6 | 95.9 | 83.8 | 96.8 |
| 16 | 93.9 | 95.9 | 83.9 | 96.7 |
| 24 | **94.2** | **96.4** | **85.4** | **96.9** |
| 32 | 93.8 | 96.0 | 84.2 | 96.7 |

Table 13: Comparison of pixel-level anomaly detection in terms of subset-wise AUROC on MVTec.

| MVTec | 1-shot SPADE | WinCLIP | PromptAD | AnomalyCLIP | Ours | 2-shot SPADE | WinCLIP | PromptAD | AnomalyCLIP | Ours | 4-shot SPADE | WinCLIP | PromptAD | AnomalyCLIP | Ours |
|---|---|---|---|---|---|---|---|---|---|---|---|---|---|---|---|
| Bottle | 66.9 | 95.1 | 94.5 | 90.4 | 98.4 | 67.3 | 95.5 | 95.9 | 90.4 | 98.6 | 67.5 | 95.2 | 96.9 | 90.4 | 98.6 |
| Cable | 57.2 | 74.6 | 76.8 | 78.9 | 94.9 | 58.1 | 76.4 | 80.7 | 78.9 | 95.3 | 60.2 | 77.1 | 83.7 | 78.9 | 96.4 |
| Capsule | 61.7 | 95.6 | 94.6 | 95.8 | 97.2 | 62.5 | 94.7 | 94.8 | 95.8 | 97.5 | 63.8 | 96.3 | 97.7 | 95.8 | 97.2 |
| Carpet | 63.1 | 99.1 | 99.1 | 98.8 | 99.5 | 62.3 | 99.1 | 99.2 | 98.8 | 99.5 | 61.8 | 99.1 | 99.2 | 98.8 | 99.5 |
| Grid | 64.7 | 95.1 | 96.8 | 97.3 | 96.6 | 66.9 | 96.0 | 96.0 | 97.3 | 96.3 | 67.6 | 96.0 | 96.8 | 97.3 | 98.1 |
| Hazelnut | 56.0 | 98.6 | 96.9 | 97.1 | 98.4 | 58.1 | 98.7 | 98.0 | 97.1 | 98.7 | 60.3 | 98.7 | 98.1 | 97.1 | 98.8 |
| Leather | 65.1 | 99.3 | 99.3 | 98.6 | 99.4 | 63.7 | 99.3 | 99.4 | 98.6 | 99.3 | 65.4 | 99.4 | 99.4 | 98.6 | 99.5 |
| Metal nut | 64.9 | 77.9 | 94.2 | 74.4 | 94.4 | 65.1 | 76.1 | 94.8 | 74.4 | 95.4 | 65.8 | 79.5 | 93.2 | 74.4 | 94.8 |
| Pill | 83.9 | 93.9 | 92.3 | 92.0 | 96.6 | 83.4 | 94.1 | 94.3 | 92.0 | 97.0 | 83.5 | 94.4 | 95.3 | 92.0 | 96.9 |
| Screw | 51.0 | 96.7 | 95.7 | 97.5 | 96.0 | 52.7 | 97.0 | 96.2 | 97.5 | 96.5 | 52.9 | 96.5 | 97.2 | 97.5 | 96.9 |
| Tile | 65.1 | 92.2 | 94.3 | 94.6 | 96.9 | 64.6 | 92.6 | 95.5 | 94.6 | 97.1 | 66.7 | 92.2 | 95.9 | 94.6 | 97.3 |
| Transistor | 58.5 | 85.1 | 75.5 | 71.0 | 86.6 | 58.9 | 95.3 | 84.4 | 71.0 | 88.4 | 59.1 | 84.4 | 87.5 | 71.0 | 89.6 |
| Toothbrush | 55.9 | 95.1 | 99.0 | 91.9 | 97.4 | 57.2 | 84.5 | 99.2 | 91.9 | 97.4 | 57.3 | 96.8 | 99.2 | 91.9 | 97.9 |
| Wood | 51.0 | 94.7 | 95.8 | 96.5 | 97.0 | 52.3 | 94.7 | 95.3 | 96.5 | 97.3 | 55.6 | 94.6 | 95.6 | 96.5 | 97.4 |
| Zipper | 41.0 | 92.9 | 97.6 | 91.4 | 96.0 | 42.4 | 92.4 | 97.3 | 91.4 | 96.7 | 46.7 | 92.9 | 96.9 | 91.4 | 96.7 |
| Mean | 60.4 | 92.4 | 91.8 | 91.1 | 96.4 | 61.2 | 92.4 | 92.2 | 91.1 | 96.7 | 62.8 | 92.9 | 92.9 | 91.1 | 97.0 |

Table 14: Comparison of pixel-level anomaly detection in terms of subset-wise PRO on MVTec.

| MVTec | 1-shot SPADE | WinCLIP | PromptAD | AnomalyCLIP | Ours | 2-shot SPADE | WinCLIP | PromptAD | AnomalyCLIP | Ours | 4-shot SPADE | WinCLIP | PromptAD | AnomalyCLIP | Ours |
|---|---|---|---|---|---|---|---|---|---|---|---|---|---|---|---|
| Bottle | 58.9 | 84.5 | 89.1 | 80.9 | 94.9 | 60.6 | 85.8 | 91.6 | 80.9 | 95.2 | 61.5 | 84.8 | 93.4 | 80.9 | 95.5 |
| Cable | 30.7 | 55.9 | 70.4 | 64.4 | 86.7 | 34.6 | 62.1 | 71.2 | 64.4 | 89.5 | 35.3 | 61.5 | 75.8 | 64.4 | 91.2 |
| Capsule | 51.4 | 89.0 | 81.2 | 87.2 | 91.3 | 52.3 | 84.2 | 82.0 | 87.2 | 92.6 | 51.7 | 88.9 | 90.7 | 87.2 | 88.9 |
| Carpet | 56.3 | 96.4 | 96.8 | 90.1 | 98.1 | 55.8 | 96.1 | 96.9 | 90.1 | 98.0 | 57.9 | 96 | 96.6 | 90.1 | 97.7 |
| Grid | 59.7 | 85.3 | 91.9 | 75.6 | 88.4 | 60.9 | 88.1 | 89.2 | 75.6 | 87.7 | 62.1 | 86.8 | 91.9 | 75.6 | 91.3 |
| Hazelnut | 47.6 | 92.5 | 90.9 | 92.4 | 93.8 | 55.6 | 93.1 | 93.6 | 92.4 | 94.6 | 54.4 | 92.4 | 93.3 | 92.4 | 95.8 |
| Leather | 61.2 | 98.2 | 98.3 | 92.2 | 98.7 | 60.7 | 98.3 | 98.8 | 92.2 | 98.5 | 61.9 | 98.2 | 97.8 | 92.2 | 98.6 |
| Metal nut | 54.6 | 77.0 | 90.1 | 71.0 | 91.8 | 57.8 | 75.6 | 90.7 | 71.0 | 92.3 | 59.9 | 79.5 | 89.4 | 71.0 | 93.4 |
| Pill | 68.5 | 88.9 | 90.1 | 58.1 | 94.8 | 66.1 | 89.6 | 93.3 | 58.1 | 94.1 | 65.3 | 89.9 | 94.6 | 58.1 | 95.6 |
| Screw | 49.8 | 87.0 | 83.9 | 88.0 | 85.3 | 50.9 | 84.1 | 84.5 | 88.0 | 85.8 | 51.2 | 86.6 | 89.1 | 88.0 | 87.5 |
| Tile | 55.1 | 78.7 | 89.8 | 87.6 | 90.4 | 54.7 | 79.7 | 90.7 | 87.6 | 90.7 | 54.6 | 78.2 | 91.4 | 87.6 | 91.2 |
| Toothbrush | 53.7 | 86.5 | 93.4 | 88.5 | 82.4 | 57.2 | 85.6 | 93.1 | 88.5 | 82.9 | 59.1 | 88.5 | 92.3 | 88.5 | 84.4 |
| Transistor | 41.9 | 63.5 | 59.4 | 58.1 | 67.4 | 43.5 | 62.8 | 66.0 | 58.1 | 68.6 | 45.3 | 62.6 | 69.4 | 58.1 | 71.2 |
| Wood | 56.7 | 86.9 | 92.8 | 91.2 | 94.2 | 58.7 | 87.8 | 93.4 | 91.2 | 94.6 | 59.5 | 88.3 | 93.1 | 91.2 | 94.7 |
| Zipper | 50.2 | 81.9 | 92.5 | 65.3 | 88.3 | 51.7 | 81.8 | 91.7 | 65.3 | 89.6 | 54.2 | 83 | 91.2 | 65.3 | 90.8 |
| Mean | 53.1 | 83.5 | 83.6 | 81.4 | 89.8 | 54.7 | 83.9 | 84.3 | 81.4 | 90.3 | 55.6 | 84.4 | 84.7 | 81.4 | 91.2 |

Table 15: Comparison of image-level anomaly detection in terms of subset-wise AUROC on MVTec.

| MVTec | 1-shot SPADE | WinCLIP | PromptAD | AnomalyCLIP | Ours | 2-shot SPADE | WinCLIP | PromptAD | AnomalyCLIP | Ours | 4-shot SPADE | WinCLIP | PromptAD | AnomalyCLIP | Ours |
|---|---|---|---|---|---|---|---|---|---|---|---|---|---|---|---|
| Bottle | 81.6 | 98.9 | 98.6 | 89.3 | 99.7 | 94.3 | 99.2 | 100.0 | 89.3 | 99.8 | 88.0 | 99.2 | 99.0 | 89.3 | 99.1 |
| Cable | 34.8 | 78.0 | 83.6 | 69.8 | 92.8 | 34.9 | 83.9 | 87.2 | 69.8 | 92.1 | 35.7 | 82.3 | 88.7 | 69.8 | 95.4 |
| Capsule | 46.5 | 75.5 | 64.2 | 89.9 | 80.5 | 65.8 | 65.5 | 65.3 | 89.9 | 91.8 | 87.4 | 80.1 | 93.4 | 89.9 | 94.5 |
| Carpet | 72.3 | 99.9 | 100 | 100.0 | 100.0 | 87.0 | 99.9 | 100.0 | 100.0 | 100.0 | 92.5 | 99.9 | 100.0 | 100.0 | 100.0 |
| Grid | 91.0 | 99.6 | 98.8 | 97.0 | 97.0 | 94.4 | 99.2 | 97.4 | 97.0 | 97.0 | 97.2 | 99.5 | 100.0 | 97.0 | 96.0 |
| Hazelnut | 49.0 | 94.9 | 98.4 | 97.2 | 98.0 | 85.3 | 95.2 | 99.8 | 97.2 | 98.5 | 92.9 | 94.7 | 99.0 | 97.2 | 98.5 |
| Leather | 61.1 | 100.0 | 100.0 | 99.8 | 100.0 | 66.2 | 100.0 | 100.0 | 99.8 | 100.0 | 74.7 | 100.0 | 100.0 | 99.8 | 100.0 |
| Metal nut | 62.2 | 98.0 | 97.6 | 93.6 | 99.4 | 70.0 | 97.8 | 96.2 | 93.6 | 99.7 | 75.5 | 98.9 | 100.0 | 93.6 | 99.9 |
| Pill | 51.8 | 88.9 | 87.9 | 81.8 | 96.6 | 60.7 | 91.8 | 89.1 | 81.8 | 96.0 | 74.7 | 91.1 | 90.4 | 81.8 | 96.6 |
| Screw | 42.4 | 85.1 | 74.0 | 81.1 | 76.8 | 48.4 | 82.7 | 81.2 | 81.1 | 81.5 | 54.8 | 84.4 | 84.2 | 81.1 | 82.1 |
| Tile | 53.7 | 100.0 | 99.8 | 100.0 | 99.7 | 55.8 | 100.0 | 99.3 | 100.0 | 99.5 | 69.8 | 100.0 | 99.2 | 100.0 | 99.9 |
| Toothbrush | 44.0 | 94.2 | 94.4 | 84.7 | 91.9 | 44.6 | 93.9 | 100.0 | 84.7 | 92.5 | 49.0 | 98.1 | 98.8 | 84.7 | 92.5 |
| Transistor | 55.0 | 85.5 | 73.7 | 92.8 | 91.4 | 55.8 | 85.4 | 87.2 | 92.8 | 90.4 | 85.6 | 85.6 | 44.4 | 92.8 | 91.2 |
| Wood | 68.8 | 98.7 | 98.6 | 96.8 | 99.4 | 74.3 | 98.9 | 98.9 | 96.8 | 99.2 | 79.4 | 98.9 | 99.2 | 96.8 | 99.6 |
| Zipper | 67.3 | 94.9 | 95.3 | 98.5 | 89.4 | 88.7 | 97.2 | 93.5 | 98.5 | 95.5 | 92.3 | 97.0 | 95.8 | 98.5 | 96.0 |
| Mean | 58.8 | 92.8 | 86.3 | 91.5 | 94.2 | 68.4 | 92.7 | 89.2 | 91.5 | 95.6 | 76.6 | 94 | 90.6 | 91.5 | 96.1 |

Table 16: Comparison of image-level anomaly detection in terms of subset-wise AUPR on MVTec.

| MVTec | 1-shot | | | | | 2-shot | | | | | 4-shot | | | | |
|---|---|---|---|---|---|---|---|---|---|---|---|---|---|---|---|
| | SPADE | WinCLIP | PromptAD | AnomalyCLIP | Ours | SPADE | WinCLIP | PromptAD | AnomalyCLIP | Ours | SPADE | WinCLIP | PromptAD | AnomalyCLIP | Ours |
| Bottle | 94.0 | 99.7 | 99.6 | 97.0 | 99.9 | 98.1 | 99.8 | 100.0 | 97.0 | 100.0 | 96.0 | 99.8 | 99.7 | 97.0 | 99.7 |
| Cable | 65.8 | 87.1 | 91.2 | 100.0 | 96.1 | 65.8 | 90.6 | 92.8 | 100.0 | 95.8 | 65.2 | 90.0 | 93.6 | 100.0 | 97.5 |
| Capsule | 76.1 | 93.9 | 85.7 | 97.9 | 95.4 | 86.8 | 88.3 | 85.5 | 97.9 | 98.3 | 95.8 | 94.8 | 98.5 | 97.9 | 98.9 |
| Carpet | 87.2 | 100.0 | 100.0 | 100.0 | 100.0 | 95.0 | 100.0 | 100.0 | 100.0 | 100.0 | 97.4 | 100.0 | 100.0 | 100.0 | 100.0 |
| Grid | 97.1 | 99.9 | 99.5 | 99.1 | 99.0 | 98.3 | 99.7 | 99.1 | 99.1 | 99.0 | 99.2 | 99.8 | 100.0 | 99.1 | 98.6 |
| Hazelnut | 80.9 | 97.4 | 99.1 | 98.6 | 99.2 | 94.4 | 97.5 | 99.9 | 98.6 | 99.4 | 97.7 | 97.2 | 99.4 | 98.6 | 99.4 |
| Leather | 74.4 | 100.0 | 100.0 | 99.9 | 100.0 | 76.7 | 100.0 | 100.0 | 99.9 | 100.0 | 83.7 | 100.0 | 100.0 | 99.9 | 100.0 |
| Metal nut | 89.7 | 99.6 | 99.3 | 98.5 | 99.9 | 91.9 | 99.5 | 98.3 | 98.5 | 99.9 | 93.7 | 99.8 | 100.0 | 98.5 | 100.0 |
| Pill | 61.5 | 97.6 | 96.8 | 95.4 | 99.4 | 71.4 | 98.3 | 97.1 | 95.4 | 99.3 | 84.5 | 98.2 | 99.7 | 95.4 | 99.4 |
| Screw | 83.2 | 95.1 | 91.19 | 92.5 | 87.0 | 85.0 | 93.3 | 93.5 | 92.5 | 92.6 | 86.6 | 94.0 | 93.7 | 92.5 | 92.0 |
| Tile | 88.1 | 100.0 | 99.9 | 100.0 | 99.9 | 88.4 | 100.0 | 99.7 | 100.0 | 99.8 | 92.4 | 100.0 | 99.6 | 100.0 | 99.9 |
| Toothbrush | 72.2 | 97.7 | 97.7 | 93.7 | 97.3 | 70.6 | 97.6 | 100.0 | 93.7 | 97.5 | 72.9 | 99.3 | 99.5 | 93.7 | 97.5 |
| Transistor | 81.1 | 80.8 | 62.2 | 90.6 | 89.8 | 73.2 | 81.0 | 77.2 | 90.6 | 88.3 | 93.5 | 82.6 | 92.2 | 90.6 | 89.2 |
| Wood | 64.5 | 99.6 | 99.5 | 99.2 | 99.8 | 70.4 | 99.7 | 99.6 | 99.2 | 99.8 | 75.6 | 99.7 | 99.7 | 99.2 | 99.9 |
| Zipper | 89.3 | 98.6 | 98.8 | 99.6 | 95.6 | 96.9 | 99.2 | 98.2 | 99.6 | 98.7 | 98.0 | 99.2 | 98.9 | 99.6 | 98.8 |
| Mean | 80.3 | 96.5 | 93.4 | 96.2 | 97.2 | 84.2 | 96.3 | 94.8 | 96.2 | 97.9 | 88.8 | 97.0 | 96.5 | 96.2 | 98.1 |

Table 17: Comparison of pixel-level anomaly detection in terms of subset-wise AUROC on VisA.

| VisA | 1-shot | | | | | 2-shot | | | | | 4-shot | | | | |
|---|---|---|---|---|---|---|---|---|---|---|---|---|---|---|---|
| | SPADE | WinCLIP | PromptAD | AnomalyCLIP | Ours | SPADE | WinCLIP | PromptAD | AnomalyCLIP | Ours | SPADE | WinCLIP | PromptAD | AnomalyCLIP | Ours |
| candle | 48.1 | 93.8 | 97.1 | 98.8 | 98.5 | 52.2 | 94.7 | 97.7 | 98.8 | 98.5 | 53.1 | 95 | 97.7 | 98.8 | 98.6 |
| capsules | 52.7 | 93.2 | 96.7 | 95.0 | 97.1 | 56.3 | 93.0 | 97.2 | 95.0 | 97.3 | 56.7 | 93.2 | 97.4 | 95.0 | 97.9 |
| cashew | 79.7 | 94.6 | 97.9 | 93.8 | 97.6 | 81.9 | 95.3 | 97.8 | 93.8 | 97.1 | 83.3 | 94.7 | 97.9 | 93.8 | 97.5 |
| chewinggum | 77.3 | 98.9 | 99.2 | 99.3 | 99.6 | 78.1 | 98.9 | 99.0 | 99.3 | 99.5 | 78.4 | 98.9 | 99.1 | 99.3 | 99.5 |
| fryum | 80.8 | 95.1 | 95.6 | 94.6 | 96.2 | 83.3 | 95.6 | 95.8 | 94.6 | 96.1 | 84.1 | 95.4 | 96.0 | 94.6 | 95.8 |
| macaroni1 | 71.7 | 95.6 | 97.6 | 98.3 | 97.5 | 71.1 | 96.7 | 98.8 | 98.3 | 97.6 | 71.5 | 97 | 98.1 | 98.3 | 97.8 |
| macaroni2 | 59.7 | 94 | 95.6 | 97.6 | 96.6 | 62.8 | 94.4 | 96.1 | 97.6 | 96.9 | 63.6 | 93.8 | 98.1 | 97.6 | 96.9 |
| pcb1 | 61.5 | 94.1 | 96.9 | 94.1 | 98.5 | 65.7 | 94.6 | 98.1 | 94.1 | 98.5 | 65.3 | 94.7 | 98.8 | 94.1 | 98.7 |
| pcb2 | 62.6 | 92.4 | 94.0 | 92.4 | 94.6 | 64.4 | 93.1 | 95.1 | 92.4 | 96.8 | 66.4 | 93.3 | 95.6 | 92.4 | 96.6 |
| pcb3 | 63.9 | 91.6 | 95.3 | 88.4 | 93 | 67.1 | 92.4 | 95.5 | 88.4 | 93.7 | 70.1 | 93.2 | 96.4 | 88.4 | 94 |
| pcb4 | 78.7 | 94.2 | 95.7 | 95.7 | 95.2 | 80.4 | 94.9 | 96.8 | 95.7 | 96.6 | 80.2 | 95.6 | 96.9 | 95.7 | 97.2 |
| pipe_fryum | 91.2 | 97.9 | 98.6 | 98.2 | 98.3 | 92.3 | 97.8 | 98.8 | 98.2 | 98.5 | 92.4 | 97.8 | 98.9 | 98.2 | 98.5 |
| Mean | 69.0 | 94.6 | 96.3 | 95.5 | 96.9 | 71.3 | 95.1 | 96.9 | 95.5 | 97.2 | 72.1 | 95.2 | 97.2 | 95.5 | 97.4 |

Table 18: Comparison of pixel-level anomaly detection in terms of subset-wise PRO on VisA.

| VisA | 1-shot | | | | | 2-shot | | | | | 4-shot | | | | |
|---|---|---|---|---|---|---|---|---|---|---|---|---|---|---|---|
| | SPADE | WinCLIP | PromptAD | AnomalyCLIP | Ours | SPADE | WinCLIP | PromptAD | AnomalyCLIP | Ours | SPADE | WinCLIP | PromptAD | AnomalyCLIP | Ours |
| candle | 65.6 | 89.6 | 92.3 | 96.2 | 95.0 | 65.4 | 90.2 | 92.3 | 96.2 | 95.3 | 65.7 | 90.5 | 92.6 | 96.2 | 95.4 |
| capsules | 51.2 | 62.1 | 82.7 | 78.5 | 83.6 | 52.9 | 61.8 | 82.1 | 78.5 | 83.9 | 53.0 | 61.9 | 77.0 | 78.5 | 85.2 |
| cashew | 59.8 | 87.6 | 89.9 | 91.6 | 92.9 | 62.3 | 86.7 | 88.1 | 91.6 | 93.0 | 63.2 | 86.7 | 88.3 | 91.6 | 92.8 |
| chewinggum | 65.9 | 82.7 | 84.9 | 91.2 | 92.5 | 68.7 | 83.0 | 84.1 | 91.2 | 93.4 | 68.9 | 82.7 | 83.2 | 91.2 | 93.7 |
| fryum | 56.5 | 87.5 | 81.9 | 86.8 | 87.7 | 54.9 | 87.8 | 80.8 | 86.8 | 88.1 | 59.9 | 88.7 | 81.9 | 86.8 | 89.2 |
| macaroni1 | 65.3 | 85.6 | 88.6 | 89.8 | 89.3 | 68.1 | 89.8 | 90.8 | 89.8 | 90.1 | 67.4 | 90.1 | 93.5 | 89.8 | 89.9 |
| macaroni2 | 61.8 | 81.0 | 83.7 | 84.2 | 86.6 | 64.6 | 81.0 | 85.2 | 84.2 | 86.3 | 65.8 | 79.8 | 91.2 | 84.2 | 87.1 |
| pcb1 | 43.2 | 68.8 | 87.9 | 81.7 | 84.3 | 47.5 | 70.2 | 86.5 | 81.7 | 85.1 | 53.1 | 70.5 | 87.1 | 81.7 | 85.6 |
| pcb2 | 46.1 | 73.6 | 73.4 | 78.9 | 77.3 | 48.0 | 74.0 | 76.8 | 78.9 | 77.8 | 51.9 | 74.1 | 77.9 | 78.9 | 77.4 |
| pcb3 | 48.7 | 76.7 | 79.0 | 77.1 | 76.8 | 52.2 | 79.4 | 79.5 | 77.1 | 78.7 | 51.1 | 80.3 | 83.6 | 77.1 | 78.1 |
| pcb4 | 49.9 | 79.9 | 76.7 | 91.3 | 83.7 | 54.1 | 82.4 | 83.7 | 91.3 | 84.3 | 54.7 | 83.8 | 82.0 | 91.3 | 87.3 |
| pipe_fryum | 72.6 | 95.7 | 96.2 | 96.8 | 97.2 | 75.9 | 95.7 | 96.9 | 96.8 | 97.1 | 76.3 | 95.8 | 96.7 | 96.8 | 97.1 |
| Mean | 57.2 | 80.9 | 82.2 | 87.0 | 87.3 | 59.6 | 81.8 | 85.2 | 87.0 | 87.9 | 60.9 | 82.1 | 84.7 | 87.0 | 88.3 |

Table 19: Comparison of image-level anomaly detection in terms of subset-wise AUROC on VisA.

| VisA | 1-shot | | | | | 2-shot | | | | | 4-shot | | | | |
|---|---|---|---|---|---|---|---|---|---|---|---|---|---|---|---|
| | SPADE | WinCLIP | PromptAD | AnomalyCLIP | Ours | SPADE | WinCLIP | PromptAD | AnomalyCLIP | Ours | SPADE | WinCLIP | PromptAD | AnomalyCLIP | Ours |
| candle | 37.3 | 96.3 | 91.8 | 79.3 | 91.9 | 57.2 | 96.4 | 92.0 | 79.3 | 95.5 | 60.0 | 96.9 | 92.9 | 79.3 | 95.9 |
| capsules | 51.8 | 79.3 | 83.2 | 81.5 | 88.9 | 56.6 | 81.6 | 78.7 | 81.5 | 90.3 | 60.1 | 83 | 81.7 | 81.5 | 90.5 |
| cashew | 64.0 | 93.9 | 88.9 | 76.3 | 85.6 | 65.1 | 92.6 | 89.6 | 76.3 | 86.7 | 80.2 | 92.6 | 88.0 | 76.3 | 91.2 |
| chewinggum | 67.5 | 97.9 | 97.3 | 97.4 | 97.7 | 74.5 | 98.1 | 97.1 | 97.4 | 97.8 | 86.6 | 98.4 | 98.1 | 97.4 | 98 |
| fryum | 72.2 | 92.8 | 88.0 | 93.0 | 89.9 | 73.9 | 90.1 | 85.7 | 93.0 | 92.7 | 83.8 | 91.6 | 90.6 | 93.0 | 93.3 |
| macaroni1 | 65.1 | 81.9 | 87.3 | 87.2 | 85.1 | 59.9 | 86.4 | 87.4 | 87.2 | 84.7 | 59.2 | 86.9 | 89.1 | 87.2 | 88.4 |
| macaroni2 | 56.0 | 78.1 | 60.8 | 73.4 | 75.5 | 50.8 | 76.8 | 74.9 | 73.4 | 76.1 | 57.2 | 79 | 80.5 | 73.4 | 78.1 |
| pcb1 | 73.4 | 83.8 | 83.0 | 85.4 | 83.5 | 78.2 | 85.5 | 82.9 | 85.4 | 86.5 | 84.1 | 86 | 86.1 | 85.4 | 85.2 |
| pcb2 | 71.7 | 58.4 | 77.9 | 62.2 | 72.6 | 79.4 | 56.8 | 84.4 | 62.2 | 75.2 | 87.2 | 59.4 | 81.1 | 62.2 | 75.5 |
| pcb3 | 58.5 | 64.9 | 79.9 | 62.7 | 71.8 | 63.7 | 67.7 | 71.7 | 62.7 | 70.7 | 74.5 | 65.6 | 87.1 | 62.7 | 74.7 |
| pcb4 | 76.6 | 72.1 | 96.5 | 93.9 | 82.9 | 89.6 | 73.6 | 96.0 | 93.9 | 84.4 | 86.9 | 70.7 | 85.3 | 93.9 | 88.7 |
| pipe_fryum | 41.7 | 98.2 | 98.9 | 92.4 | 99.8 | 53.1 | 98.5 | 99.6 | 92.4 | 99.9 | 56.6 | 98.4 | 99.3 | 92.4 | 99.8 |
| Mean | 61.3 | 83.1 | 80.8 | 82.1 | 85.4 | 66.8 | 83.7 | 84.3 | 82.1 | 86.7 | 73.0 | 84.1 | 85.7 | 82.1 | 88.3 |

Table 20: Comparison of image-level anomaly detection in terms of subset-wise AUPR on VisA.

| VisA | 1-shot | | | | | 2-shot | | | | | 4-shot | | | | |
|---|---|---|---|---|---|---|---|---|---|---|---|---|---|---|---|
| | SPADE | WinCLIP | PromptAD | AnomalyCLIP | Ours | SPADE | WinCLIP | PromptAD | AnomalyCLIP | Ours | SPADE | WinCLIP | PromptAD | AnomalyCLIP | Ours |
| candle | 41.3 | 96.7 | 90.7 | 81.1 | 94.1 | 54.6 | 96.9 | 91.7 | 81.1 | 95.3 | 57.7 | 97.3 | 92.8 | 81.1 | 95.5 |
| capsules | 65.6 | 87.0 | 90.0 | 88.7 | 94.7 | 68.8 | 89.1 | 86.7 | 88.7 | 94.9 | 71.9 | 90 | 89.0 | 88.7 | 96.0 |
| cashew | 79.4 | 97.4 | 95.0 | 89.4 | 95.7 | 80.2 | 96.7 | 95.1 | 89.4 | 95.2 | 88.2 | 96.8 | 94.7 | 89.4 | 96.0 |
| chewinggum | 83.6 | 99.1 | 98.9 | 98.9 | 99.1 | 87.2 | 99.2 | 98.8 | 98.9 | 99.2 | 93.0 | 99.3 | 99.2 | 98.9 | 99.3 |
| fryum | 86.1 | 96.9 | 94.6 | 96.8 | 96.3 | 86.9 | 95.4 | 93.9 | 96.8 | 96.8 | 91.8 | 96.1 | 95.9 | 96.8 | 97.3 |
| macaroni1 | 63.5 | 82.8 | 89.7 | 86.0 | 89.3 | 56.1 | 87.2 | 89.0 | 86.0 | 90.7 | 55.5 | 87.4 | 91.1 | 86.0 | 91.2 |
| macaroni2 | 54.1 | 80.1 | 61.5 | 72.1 | 79.3 | 51.7 | 79.0 | 78.2 | 72.1 | 80.2 | 52.0 | 81.9 | 81.2 | 72.1 | 80.6 |
| pcb1 | 73.2 | 83.7 | 77.3 | 87.0 | 77.4 | 77.7 | 84.1 | 79.0 | 87.0 | 78.5 | 84.6 | 85.6 | 81.2 | 87.0 | 78.6 |
| pcb2 | 71.6 | 58.6 | 79.3 | 64.3 | 70.2 | 78.4 | 54.6 | 85.5 | 64.3 | 73.6 | 88.5 | 61.3 | 80.8 | 64.3 | 73.9 |
| pcb3 | 58.6 | 66.2 | 81.6 | 70.0 | 73.2 | 62.1 | 67.3 | 73.5 | 70.0 | 74.3 | 75.1 | 64.6 | 88.0 | 70.0 | 74.6 |
| pcb4 | 77.5 | 73.8 | 96.1 | 94.4 | 80.9 | 88.0 | 70.2 | 94.8 | 94.4 | 81.7 | 85.7 | 73.5 | 73.6 | 94.4 | 84.5 |
| pipe_fryum | 64.2 | 99.2 | 99.6 | 96.3 | 99.9 | 72.0 | 99.3 | 99.7 | 96.3 | 100 | 75.6 | 99.3 | 99.7 | 96.3 | 99.9 |
| Mean | 68.2 | 85.1 | 83.2 | 85.4 | 87.5 | 72.0 | 84.9 | 87.8 | 85.4 | 88.6 | 76.6 | 86.1 | 88.8 | 85.4 | 89.6 |

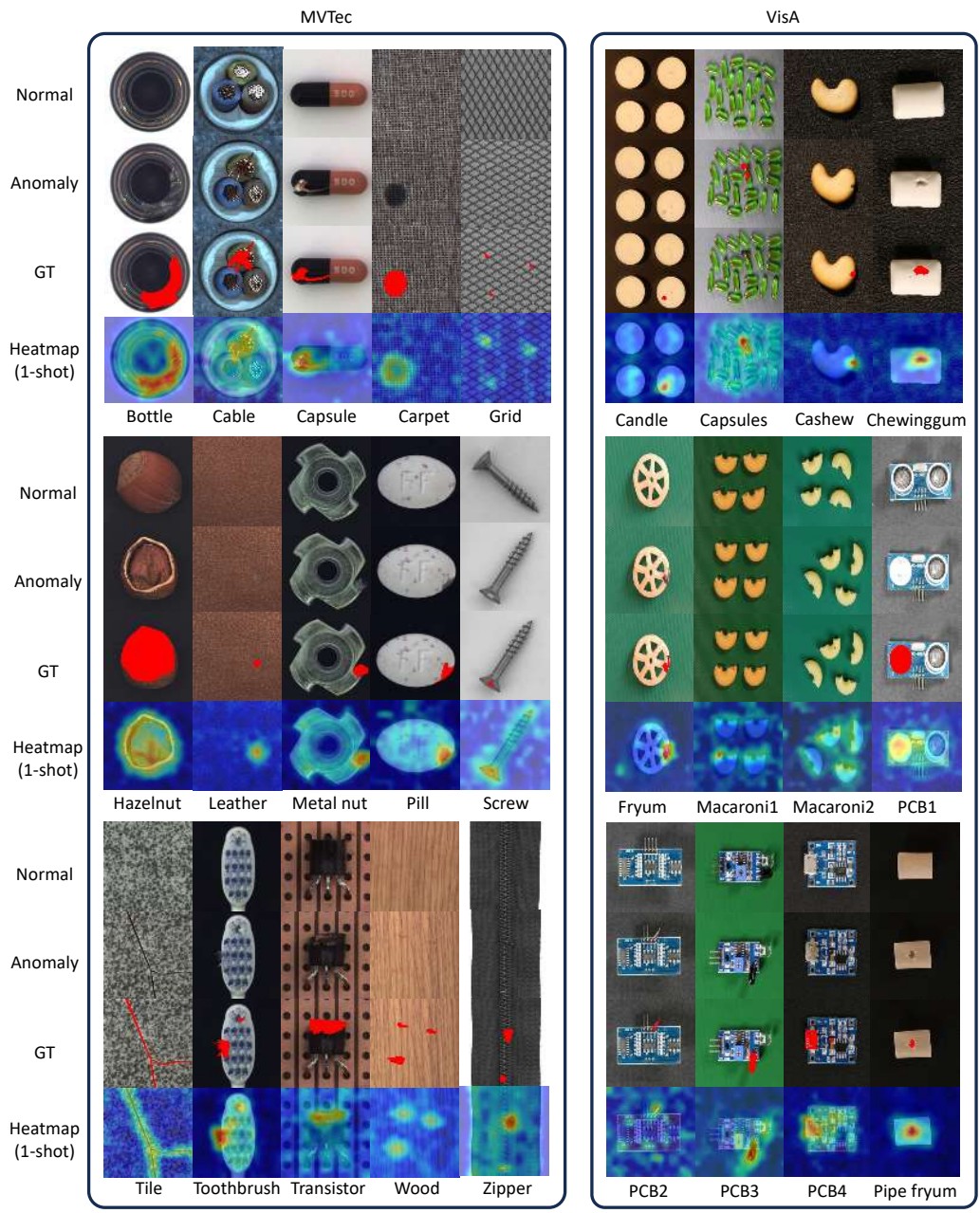

Figure 4: Additional qualitative results from our method (1-shot), tested on MVTec and VisA.

