# OpenReview forum: "One-for-All Few-Shot Anomaly Detection via Instance-Induced Prompt Learning"
_ICLR.cc/2025/Conference — ICLR 2025 Poster_

### Official Review · Reviewer_6UaR · 2024-10-21

**Soundness:** 3
**Presentation:** 2
**Contribution:** 3
**Rating:** 6
**Confidence:** 5

**Summary:**

This paper presents a new anomaly detection problem: multi-class few-shot anomaly detection (AD). The authors analyze current image-text model based few-shot anomaly detection methods such as WinCLIP, PromptAD et al. have problems in multi-class few-shot AD tasks and propose an instance-specific prompt generator, which not only improves the ability of prompt to capture abnormal regions and normal regions, but also prevents the problems caused by prompt sharing.
In addition, this paper proposes a multi-model prompt training strategy for model training, and improves the memory bank strategy for multi-category tasks to propose a class-aware memory bank.

**Strengths:**

1. This paper discusses a new and practical task.
2. The method has some novelty.
3. The experimental results show that the performance of this paper is very superior.

**Weaknesses:**

There are some problems in the organization of the paper, 1) the introduction is too redundant and not concise enough, and it is difficult to get the core views of the authors. 2) Some details of the method section are not clear enough and some parts are not reflected in Figure 1

**Questions:**

major
1. I would like the authors to briefly introduce the core insights of the paper.
2. There are many modules in the method part, and more ablation experiments are needed to demonstrate the contribution of each module, especially the instance-specific prompt generator module, in which the Projection Network and Cross Attention need to be ablated. In addition, the two losses of Synthetic Visual Guidance for Anomalous Prompt also require separate ablation experiments.
minor
3. 236 “group of patches an the embedding” “an” seems to be a misspelling
4. Table 5, I couldn't find “L_c”, if the author meant “L_s”.

---

> ### Author Response · Authors · 2024-11-24
> **Response to Reviewer 6UaR (Part1)**
>
> **Q1:** The introduction is too redundant and not concise enough, and it is difficult to get the core views of the authors. I would like the authors to briefly introduce the core insights of the paper.
>
> **A1:** We revised the introduction and shorten it to make it concise and highlight the core view. Here we briefly introduce the core insights of the paper. Current few-shot anomaly detection methods can efficiently detect and localize anomalies given a few normal samples for training, but they still follow the one-for-one paradigm, requiring training a bespoke model for every category in the dataset, which is both time- and memory-consuming. While one-for-all anomaly detection methods train a unified model to detect anomalies across different categories, but these methods require thousands of normal samples. This paper aims to propose an anomaly detection method that leverages a few normal samples from each category to train a shared model capable of detecting various anomalies across different categories in the dataset. Our method extends anomaly detection into a broader and more practical application scenario.
>
>
> **Q2:** Some details of the method section are not clear enough and some parts are not reflected in Figure 1.
>
> **A2:** We have modified Figure 1 to demonstrate our method completely. The updated Figure 1 now includes token-level alignment part of the guidance of prompt.

---

> ### Author Response · Authors · 2024-11-24
> **Response to Reviewer 6UaR (Part2)**
>
> **Q3:** There are many modules in the method part, and more ablation experiments are needed to demonstrate the contribution of each module, especially the instance-specific prompt generator module, in which the Projection Network and Cross Attention need to be ablated. In addition, the two losses of Synthetic Visual Guidance for Anomalous Prompt also require separate ablation experiments.
>
> **A3:** Thanks for raising this issue to us. To better undersand the contribution of each modules in our model, as you suggested, we further conducted additional ablation studies on the Prompt Generator and the Synthesized Visual Guidance.
>
> * First, we investigate the contribution of components in the Prompt Generator, including the projection networks, and the cross-attention module. There are two kinds of projection networks, with one responsible for producing ${\mathbf{O}}^n$ and ${\mathbf{O}}^a$, while the other one responsible for the generation of ${\mathbf{C}}^n$ and ${\mathbf{C}}^a$. Hence, we will investigate the contribution of these three components in the Prompt Generator, that is, the projection network relevant to ${\mathbf{C}}^n$ and ${\mathbf{C}}^a$, the cross-attention module that is used in producing ${\mathbf{C}}^n$ and ${\mathbf{C}}^a$, and the projection network relevant to ${\mathbf{O}}^n$ and ${\mathbf{O}}^a$. The ablation study results are shown in the table below. From the table, we can see that each component in the prompt generator contributes to the performance improving. The two projection networks are more effectiving in improving the image-level detection performance, while the cross-attention module contributes more to the improvement of pixel-level anomaly localization accuracy, indicating that the interaction between the Q-Former and VIT enhances the fine-grained description of prompts.
>
>     | Projection Network Relevant to Producing ${\mathbf{C}}^n$ and ${\mathbf{C}}^a$ | Cross Attention Relevant to Producing ${\mathbf{C}}^n$ and ${\mathbf{C}}^a$ | Projection Network Relevant to Producing ${\mathbf{O}}^n$ and ${\mathbf{O}}^a$ | Mvtec image auroc | Mvtec pixel auroc | VisA image auroc | VisA pixel auroc |
>     | ------------- | --------------- | ------------ | ----------------- | ----------------- | ---------------- | ---------------- |
>     | ×             | ×               | ×            | 93.7              | 95.9              | 84.4             | 96.3             |
>     | ×             | ×               | √            | 93.7              | 96.1              | 84.6             | 96.5             |
>     | √             | ×               | √            | 94.1              | 96.1              | 84.8             | 96.6             |
>     | ×             | √               | √            | 93.8              | 96.2              | 84.4             | 96.7             |
>     | √             | √               | √            | **94.2**          | **96.4**          | **85.4**         | **96.9**         |
>
> * Furthermore, we conducted an ablation study to investigate the impact of synthetic visual guidance. As shown in the table below, the performance of our method with synthetic visual guidance outperforms the version without them. This can be attributed to the synthetic visual guidance provides an anchor with informative content for anomalous prompts to pull closer, rather than blindly being pushed away from normal features.
>
>     | Synthetic Visual Guidance | Mvtec image auroc | Mvtec pixel auroc | VisA image auroc | VisA pixel auroc |
>     | -------------------------- | ----------------- | ----------------- | ---------------- | ---------------- |
>     | ×                          | 93.6              | 95.9              | 84.3             | 96.5             |
>     | √                          | **94.2**          | **96.4**          | **85.4**         | **96.9**         |
>
> We have added these ablation studies into the appendix of our paper.
>
> **Q4:** 236 “group of patches an the embedding” “an” seems to be a misspelling
>
> **A4:** Thank you for spotting this spelling error for us, we have revised **"an"** to **"and"**.
>
>
>
> **Q5:** Table 5, I couldn't find “L_c”, if the author meant “L_s”.
>
> **A5:** Thank you for spotting the typo in Table 5 for us, we have revised “$\mathcal{L}_c$” to “$\mathcal{L}_s$”.

---

### Official Review · Reviewer_AM88 · 2024-10-28

**Soundness:** 3
**Presentation:** 2
**Contribution:** 3
**Rating:** 6
**Confidence:** 3

**Summary:**

In this paper, the authors propose a novel "One-for-All Few-Shot" anomaly detection method, aimed at addressing the challenge of few-shot anomaly detection. Compared to traditional "One-for-One" approaches, the authors design a class-shared prompt generator that utilizes vision-language models (VLMs) to generate instance-specific prompts, improving the model's adaptability in few-shot scenarios. The method is trained to capture normality and abnormality in both visual and textual modality and introduces a category-aware memory bank to resolve the memory mismatch issue within the "One-for-All" paradigm.

**Strengths:**

1. This paper designs a novel task.
2. The proposed method is innovative and practical.
3. The experimental results show that the proposed method has achieved excellent performance on the two datasets.

**Weaknesses:**

This paper has some drawbacks and the authors can consider improving the paper's quality from the following perspectives:

1. **Grammar Mistakes and Spelling Errors**, such as:

(1) In the Abstract, it has "...fix prompts for each class, our method **learn** a class-shared prompt generator...". Here, "learn" should be in the third person singular form and changed to "...fix prompts for each class, our method **learns** a class-shared prompt generator...".

(2) In line 236, it has "we propose to encourage the alignment between a group of patches **an** the embedding of one prompt token". It seems that "an" should be replaced with "and" to ensure the correctness of the sentence.

2. **Confused Content**, such as:

In the abstract, the expression that 'We address the **mismatch problem** of the memory bank within one-for-all paradigm' is relatively ambiguous, and the specific definition of 'mismatch problem' is not clearly explained in **Sec.3 Methods**. The authors may provide more explanations about this term.

3. **Differences from others**:

Although the comparisons with PromptAD and other methods have been mentioned, there is a lack of detailed comprehensive analysis. More comparative content can be added to illustrate the specific advantages of the proposed method. For example, the author can add an independent section in the Appendix to illustrate WinCLIP/PromptAD has limitations in some specific scenarios, and how the proposed method outperforms them under these scenarios.

**Questions:**

1. **More Experimental Details**:

The impact of semantic-level alignment loss has not been extensively discussed in **Table 5**. The authors may add more experimental results to demonstrate the effectiveness of this specific loss.

2. **More Quantitative Analysis**:

The performance impact of adopting Q-Former and instance-specific prompt generator is still unexplored. For example, an extra experiment or performance analysis can be added to demonstrate the specific contribution of Q-Former to the method. My concern is that the original ViT and Q-Former are unaligned and frozen during the training, without pre-training, e.g. representation learning stage in BLIP2, they may not own the ability to align the text feature with the visual feature. Additionally, due to the CLIP having been pre-trained to align the two modality features, why the authors did not use it? In other words, the advantages of the proposed architecture are not obvious in the paper when compared with other frameworks.

3. **More theoretical analysis**:

Although the experimental results demonstrate the effectiveness of the method, the theoretical analysis is somewhat insufficient. For example, in the Prompt Generator Ablation and Loss Ablation, the authors can add one sentence to analyze and generalize why the proposed module is effective.

---

> ### Author Response · Authors · 2024-11-24
> **Response to Reviewer AM88 (Part1)**
>
> **Q1:** Grammar Mistakes and Spelling Errors.
>
>
> **A1:** Thanks a lot for spotting these grammer and spelling erros for us. We have revised them accordingly, and further examined the whole paper thoroughly to improve the writting.
>
>
>
> **Q2:** In the abstract, the expression that 'We address the **mismatch problem** of the memory bank within one-for-all paradigm' is relatively ambiguous, and the specific definition of 'mismatch problem' is not clearly explained in **Sec.3 Methods**. The authors may provide more explanations about this term.
>
> **A2:** For clarity, we revised the abstract to explicitly define the problem as 'how to retrieve valid similar features from the visual memory bank under the one-for-all paradigm.' To further eliminate the ambiguity, we provide more description and explanation of this problem in the visual memory bank. Specifically, in the one-for-all paradigm, the memory bank stores patch features from all categories. Although raw images from different categories may appear distinct, their patch features in the feature space can be close to one another. For example, wood and tile images share similar texture patterns. If the entire memory bank is used for feature matching instead of a subset specific to each category, anomalous patch features may have high similarity with normal features from totally different categories, resulting in the undesired feature match, which is called the feature mismatch problem in this paper. We have modified the section 3.3 to provide a clearer explanation to what the mismatch problem is.
>
>
> **Q3:** Although the comparisons with PromptAD and other methods have been mentioned, there is a lack of detailed comprehensive analysis. More comparative content can be added to illustrate the specific advantages of the proposed method. For example, the author can add an independent section in the Appendix to illustrate WinCLIP/PromptAD has limitations in some specific scenarios, and how the proposed method outperforms them under these scenarios.
>
> **A3:** Thank you for your advice. Our method has advantage in the scenarios which require the anomaly detection methods to follow the one-for-all paradigm. For example, we may be only able to access the raw dataset without category information in practice. In this scenario, we can not know the exact category the image belongs to, which prohibits WinCLIP and PromptAD from constructing different prompts for different categories. Our method, on the other hand, follows the one-for-all paradigm and extracts category information automatically from the image using Q-Former, thus avoiding the limitations faced by WinCLIP and PromptAD. We have added an independent section in the Appendix to introduce these specific scenarios. Moreover, the superiority of our method in such scenarios can be demonstrated by the comparison with other baselines in one-for-all paradigm as shown in Table 1.
>
>
>
> **Q4:** The impact of semantic-level alignment loss has not been extensively discussed in **Table 5**. The authors may add more experimental results to demonstrate the effectiveness of this specific loss.
>
> **A4:** As you suggested, we further conduct the ablation study to investigate the impact of semantic-level alignment loss. It can be observed that there is a significant performance drop when removing the semantic-level alignment. This is because the semantic-level alignment is the direct goal for training the prompt generator to generate suitable prompts for anomaly detection. It is consistent with the detection criterion in the test time. Therefore, semantic-level alignment loss is an essential component of our method.
>
> We have added this experiment result and analysis into the experiment section of our updated paper.
>
> | $\mathcal{L}_s$ | $\mathcal{L}_t$  | $\mathcal{L}_f$  | Mvtec image auroc | Mvtec pixel auroc | VisA image auroc | VisA pixel auroc |
> | ---- | ---- | ---- | ----------------- | ----------------- | ---------------- | ---------------- |
> | ×    | √    | √    | 92.6              | 95.6              | 83.1             | 96.2             |
> | √    | √    | √    | **94.2**          | **96.4**          | **85.4**         | **96.9**         |

---

> ### Author Response · Authors · 2024-11-24
> **Response to Reviewer AM88 (Part2)**
>
> **Q5:** The performance impact of adopting Q-Former and instance-specific prompt generator is still unexplored. For example, an extra experiment or performance analysis can be added to demonstrate the specific contribution of Q-Former to the method. My concern is that the original ViT and Q-Former are unaligned and frozen during the training, without pre-training, e.g. representation learning stage in BLIP2, they may not own the ability to align the text feature with the visual feature. Additionally, due to the CLIP having been pre-trained to align the two modality features, why the authors did not use it? In other words, the advantages of the proposed architecture are not obvious in the paper when compared with other frameworks.
>
> **A5:** Thank you for your insightful review, we will answer your questions one by one.
>
> * First, we need to point out that different from the Q-Former in BLIP-2, the Q-Former used in our method is taken from the pre-trained BILP-Diffusion model, in which the Q-Former feeds its output tokens into CLIP textual encoders and is further trained along with CLIP text encoder for alignment. Thus, the Q-Former's output tokens ${\mathbf{Z}}$ in BILP-Diffusion are well aligned with the input space of CLIP textual encoder, i.e., the textual space. So, the Q-Former tokens ${\mathbf{Z}}$ can be roughtly viewed as the word tokens, and thus can be input to the CLIP textual encoder to output vision-aligned representations.
>
> * Second, to explain why we use Q-Former instead of CLIP, we first need to clarify that in our model, Q-Former's tokens will be fed into CLIP's textual encoder to generate the embeddings ${\mathbf{e}}_i^n$ and ${\mathbf{e}}_i^a$. This is feasible because the Q-Former in BLIP-Diffusion is trained to output tokens that not only describe the image content but also can be interpreted by the CLIP textual encoder. That is, tokens output from Q-Former can be roughly viewed as word tokens. Instead, tokens from CLIP visual encoder are trained to be aligned with the CLIP's textual encoder's output, rather than  aligned with the input space of CLIP's textual encoder. Thus, we canot simply feed tokens from CLIP's visual encoder into CLIP's textual encoder. That's one of the reasons why we choose to use the Q-Former in our model.
> In addition to this reason, we also want to emphasize that tokens output from Q-Former are much more informative than the token from CLIP's visual encoder. That's because Q-Former is trained to output multiple tokens that can describe the image content, while CLIP is only trained to output a CLS token that is  aligned with the textual representation. Obviously, in the visual anomaly detection task, we expect the tokens contain as much information of image as possible. This is also a reason why we use Q-Former instead of CLIP.
> To verify the arguments elaborated above, we conduct an additional ablation study to investigate the impact of replacing Q-Former with the CLIP visual encoder. The results are shown in the table below, from which we can see a significant performance drop when the CLIP visual encoder is used.
>
>     | CLIP Visual Encoder CLS Tokens | Q-Former Tokens | Mvtec image auroc | Mvtec pixel auroc | VisA image auroc | VisA pixel auroc |
>     | ------------------------------- | -------------------------------- | ----------------- | ----------------- | ---------------- | ---------------- |
>     | √                               | ×                                | 84.4              | 88.5              | 73.1             | 90.1             |
>     | ×                               | √                                | **94.2**          | **96.4**          | **85.4**         | **96.9**         |
>
>
> * Third, to see the impact of the instance-specific prompt generator, we conduct an experiment to compare the performance of methods that use the generated instance-specific prompts ${\mathbf{S}}_i^n$ and ${\mathbf{S}}_i^a$ or the simple instance-shared prompts, that is, removing the instance-specific tokens $\mathbf{O}^n_i$, $\mathbf{O}^a_i$ and $\mathbf{C}^n_i$, $\mathbf{C}^a_i$ from ${\mathbf{S}}_i^n$ and ${\mathbf{S}}_i^a$. The results are shown in the table below, from which we can clearly see the advantage of using the proposed instance-specific prompts.
>
>     | Instance-Specific Prompt | Instance-Shared Prompt | Mvtec image auroc | Mvtec pixel auroc | VisA image auroc | VisA pixel auroc |
>     | ------------------------- | ---------------------- | ----------------- | ----------------- | ---------------- | ---------------- |
>     | ×                         | √                      | 90.6              | 95.8              | 83.7             | 96.2             |
>     | √                         | ×                      | **94.2**          | **96.4**          | **85.4**         | **96.9**         |

---

> ### Author Response · Authors · 2024-11-24
> **Response to Reviewer AM88 (Part3)**
>
> **Q6:** Although the experimental results demonstrate the effectiveness of the method, the theoretical analysis is somewhat insufficient. For example, in the Prompt Generator Ablation and Loss Ablation, the authors can add one sentence to analyze and generalize why the proposed module is effective.
>
> **A6:** Thank you for your advice. We have revised the Prompt Generator Ablation and Loss Ablation and provide more analyses of proposed modules.
> * For prompt generator ablation study, the effectiveness of normal and anomalous tokens $\mathbf{O}^n$ and $\mathbf{O}^a$ that are appended at the end of generated prompts can be attributed to the fact that these tokens provide detailed object information of normality and abnormality. Incorporating the instance-specific visual tokens $\mathbf{C}^n$ and $\mathbf{C}^a$ is beneficial since they provide more instance-specific visual information.
> * For loss ablation study, we analyze that the semantic alignment loss $\mathcal{L}_s$, which is aligned with the detection behavior during testing, serves as the primary objective for training the prompt generator. The token-level alignment loss $\mathcal{L}_f$ encourages the prompt tokens to focus on specific patches, enhancing the anomaly localization capability of the generated prompts. Meanwhile, the textual guidance loss $\mathcal{L}_t$ incorporates expert knowledge from manually crafted text prompts into the prompt generator, enabling it to produce prompts that capture real normality and abnormality.
>
> We have revised the content of ablation studies in the updated version.

---

### Official Review · Reviewer_9Lmw · 2024-10-28

**Soundness:** 3
**Presentation:** 2
**Contribution:** 3
**Rating:** 8
**Confidence:** 3

**Summary:**

This paper proposes a novel anomaly detection methodology for industrial applications in a one-for-all categories paradigm. It uses prompt tuning and contrastive learning to pull training images and normal prompts closer together and multi-level fusion to create pseudo-anomalies which are pulled together with the anomaly prompts.

**Strengths:**

The methodology appears to be sound and introduces several steps.

The experimentation is comprehensive and the results show consistent improvement over baselines.

The paper is mostly well written and clear.

**Weaknesses:**

Please see questions

**Questions:**

1. On line 218, how are the selected subset of F for prompt learning chosen?

2. It is not clear to me why building the category-aware memory bank using image patch tokens as well as category tokens is necessarily better than the memory banks used in previous methods. Is it the case that query samples from one category were often being matched to memory bank items from other categories? This seems unlikely.

3. How does the additional training and prompt tuning affect computation complexity / runtime compared with the base methods?

---

> ### Author Response · Authors · 2024-11-24
> **Response to Reviewer 9Lmw**
>
> Thank you for your positive and insightful review, here is our response:
>
> **Q1:** how are the selected subset of F for prompt learning chosen?
>
> **A1:** We select patch visual embedding outputs from the 6-th, 12-th, 18-th, and 24-th layers of the CLIP visual encoder VIT-L/14, which contain the visual information from low-level to high-level, to form the selected subset $\hat{\mathcal{F}}_{i}$. This selection can help prompt generator to learn more general and comprehensive normality and abnormality. We have mentioned this implementation detail in the Appendix of our paper.
>
>
>
> **Q2:** It is not clear to me why building the category-aware memory bank using image patch tokens as well as category tokens is necessarily better than the memory banks used in previous methods. Is it the case that query samples from one category were often being matched to memory bank items from other categories? This seems unlikely.
>
> **A2:** Under the one-for-all paradigm, the memory bank stores patch features from all categories. Despite images may come from different categories, their patch features may be close in the feature space. For example, wood and tile images can share similar texture patterns. Since we don't have the category name of the testing and training images under the one-for-all paradigm, if the vanilla memory bank used in previous methods is employed, an anomalous patch of a testing image is probably found to be similar with a normal patch from a different category in the memory bank, resulting the missing detection. Thus, under the one-for-all paradigm, it is important to first predict the possible category of an image, and then only searching the similar patches with the same category in the memory bank. As shown in Table 6, the performance of a category-aware memory bank surpasses that of the vanilla memory bank, indicating the presence of the mismatch issue for the vanilla memory bank under the one-for-all paradigm and demonstrating that our category-aware memory bank can effectively address this problem.
>
>
> **Q3:** How does the additional training and prompt tuning affect computation complexity / runtime compared with the base methods?
>
> **A3:** Since we train a shared and lightweight prompt generator for all categories in the dataset, the training computational complexity of our method is lower than that of methods following the one-for-one paradigm. There is no need to tune bespoke prompts for different categories, which can be time-consuming. Under the one-shot setting, our method trains the prompt generator for 3 epochs to achieve optimal performance at the pixel level and 5 epochs at the image level. Moreover, during testing, we do not need to divide a raw image into hundreds of windows of different sizes for feature extraction as WinCLIP does. The runtime performance of our method during testing is competitive with the base methods.

---

### Official Review · Reviewer_Nv81 · 2024-11-03

**Soundness:** 3
**Presentation:** 3
**Contribution:** 3
**Rating:** 6
**Confidence:** 3

**Summary:**

This study proposes an instance-specific prompt generator and a category-aware memory bank aligned with a new one-for-all paradigm in few-shot anomaly detection.

**Strengths:**

This study proposes a novel prompt generation method, utilizing a class-aware memory bank to store visual features by class and extract normal and abnormal features tailored to each instance. As a result, it achieved the highest performance in the new one-for-all task.

**Weaknesses:**

- In the overall performance comparison table supporting the proposed methodology, the performance of "one-for-all" and "one-for-one" is identical, yet no interpretation is provided for this outcome.
- In Figure 1, the representation of P is omitted.
- The meaning of the output values of the newly applied Q-Former in this study is not explained.
- In Equation 12, there is an undefined loss term.
- In Table 4, it is essential to confirm whether variables outside the experimental modules were well controlled. Given that the highest performance was achieved by adding the M1 and M2 modules, it seems possible that other modules were incorporated at intermediate stages.
- Table 5 lacks definitions and contains errors in the loss expressions, making comparison and evaluation challenging.
- In Table 6, the AUROC performance for VisA at the image level differs, indicating a need to verify whether the experiments were conducted accurately.

**Questions:**

1. Given that both normal and anomalous object tokens 𝑂 are output from the same MLP, please clarify whether they utilize the same learnable token. Additionally, if the same token is used, it would be helpful to explain how the MLP architecture enables the generation of distinct normal and anomalous tokens.
2. To validate the claim that Gaussian noise is not required, consider including an experiment comparing the proposed method to a version that incorporates Gaussian noise. Such a comparison could illustrate the impact of Gaussian noise on performance and provide evidence for this design choice.
3. In Table 4, the shift from OVB to CAMV in the Memory Bank module when adding M2 could explain some performance improvements. To better understand each module's role, an additional ablation study that isolates the effects of the Prompt Generator and the Class-aware Memory Bank would clarify their individual contributions to performance in the one-for-all task.
4. The identical performance results in Tables 1 and 2 between the class-wise training in the one-for-one setting and the one-for-all setting are intriguing. A more detailed explanation of how the method differs across these settings, as well as a discussion on why the performance remains the same, would provide valuable insights into the implications for the method's contributions.

---

> ### Author Response · Authors · 2024-11-24
> **Response to Reviewer Nv81 (Part1)**
>
> **Q1:** In the overall performance comparison table supporting the proposed methodology, the performance of "one-for-all" and "one-for-one" is identical, yet no interpretation is provided for this outcome.
>
> **A1:** In Table 2, our method remains the one-for-all paradigm, as explained in the table caption, to demonstrate that the performance of our method is also competitive with baselines in the one-for-one paradigm. Therefore, the performance of our method is identical in both Table 1 and Table 2. To eliminate the ambiguity, we modified the notation of our method in Table 2 from 'ours' to 'ours(one-for-all)'.
>
> **Q2:** In Figure 1, the representation of P is omitted.
>
> **A2:** We have updated Figure 1 to illustrate more details of our model, including how to obtain the omitted representation of ${\mathbf{p}}_{ij}$. In addition, we further explicitly add the module that shows how to realize the token-level alignment.
>
>
>
> **Q3:** The meaning of the output values of the newly applied Q-Former in this study is not explained.
>
> **A3:** The Q-Former used in this paper is adopted from the BLIP-Diffusion, which takes in images and is trained to output a set of tokens that can describe the image content. Since Q-Former is connected to the input of CLIP’s textual encoder and is then jointly trained with CLIP in BLIP-Diffusion, it has been observed that the output tokens from Q-Former can be roughly viewed as word tokens that describe the image content. That's why we propose to use tokens output from Q-Former to construct prompts and then pass the prompts into the CLIP textual encoder.
>
> To make readers understand the Q-Former tokens more easily, we have added the explanations above into the original paper.
>
>
> **Q4:** In Equation 12, there is an undefined loss term.
>
> **A4:** Thanks for spotting this missing for us. The two undefined losses $\tilde{\mathcal{L}}^n_f$ and $\tilde{\mathcal{L}}^a_f$ in Eq. (12) means of the average of $\mathcal{L}^n_f$ and $\mathcal{L}^a_f$ over the selective feature subset $\hat{\mathcal{F}}_i$. In our new version, to make readers  understand this content more easily, we absorb the average operation into the  losses $\mathcal{L}^n_f$ and $\mathcal{L}^a_f$ in Eq. (7) and (10), and thus don't need to define the losses $\tilde{\mathcal{L}}^n_f$ and $\tilde{\mathcal{L}}^a_f$ anymore. In the new version, the training loss Eq. (12) can be directly written as $\mathcal{L} = (\mathcal{L}^n_s + \mathcal{L}^a_s) + \beta ({\mathcal{L}^n_f + \mathcal{L}^a_f}) + \alpha\mathcal{L}_t$, which does not need the undefined loss terms $\tilde{\mathcal{L}}^n_f$ and $\tilde{\mathcal{L}}^a_f$.
>
> **Q5:** In Table 4, it is essential to confirm whether variables outside the experimental modules were well controlled. Given that the highest performance was achieved by adding the M1 and M2 modules, it seems possible that other modules were incorporated at intermediate stages.
>
> **A5:** We are sorry for the ambiguity introduced. We have completely revised the statement regarding this point in our paper to  elaborate how this ablation study is performed. Actually, the component $M_1$ here refers to whether adding the instance-specific visual tokens $\mathbf{C}^n, \mathbf{C}^a$ into the designed prompts ${\mathbf{S}}_i^n$ and ${\mathbf{S}}_i^a$, while $M_2$ refers to whether appending the tokens $\mathbf{O}^n$,  $\mathbf{O}^a$ into the prompts. We can clearly see that the two types of tokens could be incorporated into the prompts independently, without affecting the implementation of other components of the model. Thus, we can evaluate the contribution of the two components $M_1$ and $M_2$ to the performance individually, without affecting the other modules in the model. We want to emphasize that when we perform this ablation study, all the other modules in the model are kept intact.
>
> **Q6:** Table 5 lacks definitions and contains errors in the loss expressions, making comparison and evaluation challenging.
>
> **A6:** Thanks for spotting the typos in Table 5 for us. The term ${\mathcal{L}}_c$ in the original table should be ${\mathcal{L}}_s$, which represents the semantic-level visual guidance loss. We have corrected it in our revised version. The definitions of the different loss notations in Table 5 have also been added to the loss ablation study for clarity.

---

> ### Author Response · Authors · 2024-11-24
> **Response to Reviewer Nv81 (Part2)**
>
> **Q7:** In Table 6, the AUROC performance for VisA at the image level differs, indicating a need to verify whether the experiments were conducted accurately.
>
> **A7:** Thank you for your careful reading and spotting this mistake for us. We mistakenly wrote the pixel-level PRO performance into the cell of image-level AUROC when we transferred the experimental results from the local file to the LaTeX document. This is corroborated by the identity between our originally reported performance `87.3` in Table 6 and the PRO performance of VisA under the one-shot setting in Table 1. In our new version, we have updated this number in Table 6 from `87.3` into 85.4 accordingly.
>
>
>
> **Q8:** Given that both normal and anomalous object tokens ${\mathbf{O}}$ are output from the same MLP, please clarify whether they utilize the same learnable token. Additionally, if the same token is used, it would be helpful to explain how the MLP architecture enables the generation of distinct normal and anomalous tokens.
>
> **A8:** Sorry for the ambiguity introduced. In our method, the two tokens ${\mathbf{O}}^n$ and ${\mathbf{O}}^a$ are derived from the same token, that is, the Q-Former output tokens ${\mathbf{Z}}\in{\mathbb{R}}^{M\times d}$. The reason why we can obtain two distinct tokens ${\mathbf{O}}^n$ and ${\mathbf{O}}^a$ is that we employed two MLPs here to perform the mapping. Initially, the two MLPs can only ensure the obtained tokens are distinct. But as the training proceeds, the training loss in Eq. (12) will encourage the two MLPs to output tokens that describe the normality and abnormality in images, respectively.
>
>
>
>
>
> **Q9:** To validate the claim that Gaussian noise is not required, consider including an experiment comparing the proposed method to a version that incorporates Gaussian noise. Such a comparison could illustrate the impact of Gaussian noise on performance and provide evidence for this design choice.
>
> **A9:** As you suggested, we further conduct a ablation study to investigate the impact of Gaussian noise on synthetic visual feature, with the results shown in the table below. From the table, we can see a significant performance drop when the Gaussian noise is removed from the synthetic visual feature. This can be attributed to the role of Gaussian noise in smoothing the synthetic visual feature and pushing it into low-density areas. Notably, SimpleNet [1] also incorporates Gaussian noise into their normal features to construct pseudo-anomalous features. Due to the limitation of space, we have added this ablation results into the appendix of our paper.
>
>
> | Gaussian Noise | Mvtec image auroc | Mvtec pixel auroc | VisA image auroc | VisA pixel auroc |
> | -------------- | ----------------- | ----------------- | ---------------- | ---------------- |
> | ×              | 92.5              | 95.4              | 83.1             | 96.4             |
> | √              | **94.2**          | **96.4**          | **85.4**         | **96.9**         |
>
> [1] Liu Z, Zhou Y, Xu Y, et al. Simplenet: A simple network for image anomaly detection and localization CVPR 2023.
>
> **Q10:** In Table 4, the shift from OVB to CAVB in the Memory Bank module when adding M2 could explain some performance improvements. To better understand each module's role, an additional ablation study that isolates the effects of the Prompt Generator and the Class-aware Memory Bank would clarify their individual contributions to performance in the one-for-all task.
>
> **A10:** As stated in the reply to Q5, when we perform the the ablation study on the contributions of $M_1$ and $M_2$ in the Prompt Generator, we only change the forms of the prompts ${\mathbf{S}}_i^n$ and ${\mathbf{S}}_i^a$, i.e., whether including the tokens $\mathbf{C}^n, \mathbf{C}^a$ or tokens $\mathbf{O}^n$, $\mathbf{O}^a$ into the prompts, while keeping all other modules in the model intact, including the category-aware memory bank module CAVB. Thus, the results in Table 4 have already reflected the effects of individual components ($M_1$ and $M_2$) in the Prompt Generator, while frozing the other modules. Here,  $M_1$ indicates whether adding the tokens $\mathbf{C}^n, \mathbf{C}^a$ into the prompts ${\mathbf{S}}_i^n$ and ${\mathbf{S}}_i^a$, while $M_2$ indicating whether appending $\mathbf{O}^n$, $\mathbf{O}^a$ at the end of the prompts. Please refer to the reply to Q5 for more details.
>
> Similarly, when performing the ablation study on the effect of each component in the memory bank, we also kept all other modules in the model intact. Thus, results in Table 6 have already reflected the impact of class-aware memory bank when having all other modules frozen.

---

> ### Author Response · Authors · 2024-11-24
> **Response to Reviewer Nv81 (Part3)**
>
> **Q11:** The identical performance results in Tables 1 and 2 between the class-wise training in the one-for-one setting and the one-for-all setting are intriguing. A more detailed explanation of how the method differs across these settings, as well as a discussion on why the performance remains the same, would provide valuable insights into the implications for the method's contributions.
>
> **A11:** As elaborated in the caption of Table 2, our method in this table is still the one-for-all method, although all other methods in the table are the one-for-one method. This explains why  identical performance of our method is observed in Table 1 and 2. The reason why we compare our method belonging to the one-for-all paradigm with the other one-for-one methods is to demonstrate the superiority of our proposed one-for-all method, even when comparing it with one-for-one methods. To make the one-for-all characteristic of our method more notable, we have updated the name of our method in Table 2 from 'ours' to 'ours(one-for-all)' in the new version of our paper.

---

> ### Author Response · Authors · 2024-12-01
> **Do our responses addressed your concerns?**
>
> Thanks again for your detailed and insightful comments. We have presented very detailed comments to every question you raised. In addition, we also revised our paper to improve the presentation clarity thoroughly. We hope our responses and revisions to the paper can address your concerns. If you still have any further questions, please let us know, and we will try our best to explain it to you.

---

> > ### Comment · Reviewer_Nv81 · 2024-12-01
> >
> > Thank you for providing a detailed response to my questions. Your explanation addressed the concerns. I will update my evaluation to reflect a more positive assessment of your paper, leaning towards an accept decision.

---

> ### Author Response · Authors · 2024-12-01
>
> We really appreciate your reply and the decision to raise your rating!

---

### Official Review · Reviewer_epHH · 2024-11-05

**Soundness:** 3
**Presentation:** 3
**Contribution:** 3
**Rating:** 6
**Confidence:** 5

**Summary:**

This paper introduces a novel anomaly detection challenge: multi-class few-shot anomaly detection (AD). The authors examine the current few-shot AD methods using image-text models, such as WinCLIP and PromptAD, and identify the key limitations when applied to multi-class few-shot AD tasks. In response, they propose an instance-specific prompt generator that enhances the prompt's capacity to identify both anomalous and normal regions while mitigating issues that arise from shared prompts. Additionally, the paper introduces a multi-model prompt training strategy to strengthen model training and modifies the memory bank approach for multi-class tasks by proposing a class-aware memory bank.

**Strengths:**

1.The authors have innovatively proposed a new detection task that meets the practical industrial needs. They tested numerous outstanding anomaly detection algorithms on this task.
2.The proposed anomaly detection algorithm performs well in multi-class anomaly detection within few-shot scenarios. This method incorporates the capabilities of several currently popular large models. The proposed prompt learning strategy is innovative.

**Weaknesses:**

1. The paper contains several issues that affect its clarity and focus. The introduction covers too broad a range of topics and fails to highlight the core theme of the article. Additionally, Figure 1 does not fully and clearly illustrate the methodology of this paper, especially the Guidance of Prompt Learning section. It also fails to adequately represent the complex visual guidance process described in Section 3.2.1. Furthermore, there is a grammatical error with the second 'S's in line 203 of the paper.
2. The method contains too many modules, and the ablation experiments are insufficient to demonstrate whether the certain modules  contribute to the final experimental results, especially in the Prompt Generator Ablation part. The existing ablation experiments do not clearly show whether the projection network and cross-attention are truly useful. The method uses too many loss functions, and the analysis in the ablation experiments is not accurate enough. For example, the role of Synthetize Visual Features is not clearly stated.
3. AnomalyClip and InCTRL are representative works in the same field but this paper does not compare their performance in the experiments section.

**Questions:**

Please kindly refer to the weakness.

---

> ### Author Response · Authors · 2024-11-24
> **Response to Reviewer epHH (Part1)**
>
> **Q1:** The introduction covers too broad a range of topics and fails to highlight the core theme of the article. Figure 1 does not fully and clearly illustrate the methodology of this paper. There is a grammatical error with the second 'S's in line 203 of the paper.
>
> **A1:**
> 1) Thanks for pointing out the issue on Introduction to us. We have rewrite the Introduction and shorten it to make it more concise and highlight the core theme of our paper.
> 2) As for the issue of Figure 1, we have redraw the figure to provide more details to illustrate the model more clearly.
> 3) Thank you for the careful reading. We have revised the $\mathbf{S}^n_a$ to $\mathbf{S}^a_i$.
>
>
> **Q2:** The method contains too many modules, and the ablation experiments are insufficient to demonstrate whether the certain modules contribute to the final experimental results, especially in the Prompt Generator Ablation part. The existing ablation experiments do not clearly show whether the projection network and cross-attention are truly useful. The method uses too many loss functions, and the analysis in the ablation experiments is not accurate enough. For example, the role of Synthetize Visual Features is not clearly stated.
>
> **A2:** Thanks for raising this issue to us. To better undersand the contribution of each component in our model, as you suggested, additional ablation studies were further conducted on the Prompt Generator and the Synthesized Visual Features.
>
> * First, we investigate the contribution of components in the Prompt Generator, including the projection networks, and the cross-attention module. There are two kinds of projection networks, with one responsible for producing ${\mathbf{O}}^n$ and ${\mathbf{O}}^a$, while the other one responsible for the generation of ${\mathbf{C}}^n$ and ${\mathbf{C}}^a$. Hence, we will investigate the contribution of these three components in the Prompt Generator, that is, the projection network relevant to ${\mathbf{C}}^n$ and ${\mathbf{C}}^a$, the cross-attention module that is used in producing ${\mathbf{C}}^n$ and ${\mathbf{C}}^a$, and the projection network relevant to ${\mathbf{O}}^n$ and ${\mathbf{O}}^a$. The ablation study results are shown in the table below. From the table, we can see that each component in the prompt generator contributes to the performance improving. The two projection networks are more effectiving in improving the image-level detection performance, while the cross-attention module contributes more to the improvement of pixel-level anomaly localization accuracy, indicating that the interaction between the Q-Former and VIT enhances the fine-grained description of prompts.
>
>     | Projection Network Relevant to Producing ${\mathbf{C}}^n$ and ${\mathbf{C}}^a$ | Cross Attention Relevant to Producing ${\mathbf{C}}^n$ and ${\mathbf{C}}^a$ | Projection Network Relevant to Producing ${\mathbf{O}}^n$ and ${\mathbf{O}}^a$ | Mvtec image auroc | Mvtec pixel auroc | VisA image auroc | VisA pixel auroc |
>     | ------------- | --------------- | ------------ | ----------------- | ----------------- | ---------------- | ---------------- |
>     | ×             | ×               | ×            | 93.7              | 95.9              | 84.4             | 96.3             |
>     | ×             | ×               | √            | 93.7              | 96.1              | 84.6             | 96.5             |
>     | √             | ×               | √            | 94.1              | 96.1              | 84.8             | 96.6             |
>     | ×             | √               | √            | 93.8              | 96.2              | 84.4             | 96.7             |
>     | √             | √               | √            | **94.2**          | **96.4**          | **85.4**         | **96.9**         |
>
> * For the loss function, we have revised this part and added more analyses on each loss in our paper. Furthermore, we conducted another ablation study to investigat the effectiveness of synthetic visual features. As shown in the table below, by using the synthetic visual features, the performance of our method can be improved substantially. The improvement is probably because synthetic visual features can serve as an anchor that guides the learning of anomalous prompts, rather than blindly pushing anomalous prompts away from normal features.
>
>     | Synthetic Visual Features | Mvtec image auroc | Mvtec pixel auroc | VisA image auroc | VisA pixel auroc |
>     | ------------------------- | ----------------- | ----------------- | ---------------- | ---------------- |
>     | ×                         | 93.6              | 95.9              | 84.3             | 96.5             |
>     | √                         | **94.2**          | **96.4**          | **85.4**         | **96.9**         |
>
> These additional ablation studies shown above have been added into the appendix of our paper.

---

> ### Author Response · Authors · 2024-11-24
> **Response to Reviewer epHH (Part2)**
>
> **Q3:** AnomalyClip and InCTRL are representative works in the same field but this paper does not compare their performance in the experiments section.
>
> **A3:** Thanks for this question. Actually, in Table 1 of our original paper, the comparison with the zero-shot method AnomalyClip is already included. The reason why we didn't compare with InCTRL is that the InCTRL paper only targeted at the image-level anomaly detection task, but didn't conduct the pixel-level anomaly localization. We agree it is better to include the comparison with this very relevant method. Hence, here we take the reported results from the original InCTRL paper and compare them with ours, as shown in the table below. Since the original InCTRL paper only reports the Image-level detection performance under 2, 4 and 8-shot cases, we only show the comparison under these settings. From the table, we can see that our method outperforms InCTRL the these scenarios.
>
> In our updated version, we have included the comparison with InCTRL in the Table 1.
>
>
> | Setting | Method | Mvtec  AUROC$_I$ | Mvtec AUPR | Mvtec  AUROC$_p$ | Mvtec PRO | VIsA AUROC$_I$ | VisA AUPR | VIsA AUROC$_p$ | VisA PRO |
> | ------- | ------ | ---------------- | ---------- | ---------------- | --------- | -------------- | --------- | -------------- | -------- |
> | 1shot   | InCTRL | -                | -          | -                | .         | -              | -         | -              | -        |
> |         | Ours   | **94.2**         | **97.2**   | **96.4**         | **89.8**  | **85.4**       | **87.5**  | **96.9**       | **87.3** |
> | 2shot   | InCTRL | 94.0             | 96.9       | -                | -         | 85.8           | 87.7      | -              | -        |
> |         | Ours   | **95.7**         | **97.9**   | **96.7**         | **90.3**  | **86.7**       | **88.6**  | **97.2**       | **87.9** |
> | 4shot   | InCTRL | 94.5             | 97.2       | -                | -         | 87.7           | **90.2**  | -              | -        |
> |         | Ours   | **96.1**         | **98.1**   | **97.0**         | **91.2**  | **88.3**       | 89.6      | **97.4**       | **88.3** |

---

### Meta-Review · Area_Chair_2cCk · 2024-12-11

**Metareview:**

This paper introduces a novel few-shot anomaly detection approach and extensive experiments demonstrate significant performance gains on benchmarks like MVTec and VisA, with ablation studies validating the effectiveness of each module. While initial concerns from reviewers included clarity of the introduction, insufficient ablation studies, unclear terminology, and missing comparisons with methods like InCTRL, the authors addressed these issues during the rebuttal period by revising the introduction, adding new experiments, and providing detailed explanations. All reviewers gave positive scores in the end.

**Additional Comments On Reviewer Discussion:**

During the rebuttal period, the authors addressed key concerns from reviewers including the clarify of the presentation and experimental comparisons. The authors' proactive responses to revise the paper substantially influenced the final decision to accept.

---

### Decision · Program_Chairs · 2025-01-22

Accept (Poster)